# An expanded allosteric network in PTP1B by multitemperature crystallography, fragment screening, and covalent tethering

Daniel A Keedy[1†‡§#], Zachary B Hill[2†], Justin T Biel[1], Emily Kang[2], T Justin Rettenmaier[3¶], José Brandão-Neto[4], Nicholas M Pearce[5], Frank von Delft[4,6,7], James A Wells[2,3], James S Fraser[1*]

[1]Department of Bioengineering and Therapeutic Sciences, University of California, San Francisco, San Francisco, United States; [2]Department of Pharmaceutical Chemistry, University of California, San Francisco, San Francisco, United States; [3]Cellular and Molecular Pharmacology, University of California, San Francisco, San Francisco, United States; [4]Diamond Light Source, Didcot, United Kingdom; [5]Crystal and Structural Chemistry Group, Bijvoet Center for Biomolecular Research, Utrecht University, Utrecht, Netherlands; [6]Structural Genomics Consortium, University of Oxford, Oxford, United Kingdom; [7]Department of Biochemistry, University of Johannesburg, Johannesburg, South Africa

*For correspondence:
jfraser@fraserlab.com

[†]These authors contributed equally to this work

Present address: [‡]Structural Biology Initiative, CUNY Advanced Science Research Center, New York, United States; [§]Department of Chemistry and Biochemistry, City College of New York, New York, United States; [#]Biochemistry and Chemistry PhD Programs, Graduate Center, City University of New York, New York, United States; [¶]Jnana Therapeutics, Cambridge, United States

Competing interests: The authors declare that no competing interests exist.

**Abstract** Allostery is an inherent feature of proteins, but it remains challenging to reveal the mechanisms by which allosteric signals propagate. A clearer understanding of this intrinsic circuitry would afford new opportunities to modulate protein function. Here, we have identified allosteric sites in protein tyrosine phosphatase 1B (PTP1B) by combining multiple-temperature X-ray crystallography experiments and structure determination from hundreds of individual small-molecule fragment soaks. New modeling approaches reveal 'hidden' low-occupancy conformational states for protein and ligands. Our results converge on allosteric sites that are conformationally coupled to the active-site WPD loop and are hotspots for fragment binding. Targeting one of these sites with covalently tethered molecules or mutations allosterically inhibits enzyme activity. Overall, this work demonstrates how the ensemble nature of macromolecular structure, revealed here by multitemperature crystallography, can elucidate allosteric mechanisms and open new doors for long-range control of protein function.

DOI: https://doi.org/10.7554/eLife.36307.001

## Introduction

Proteins are collections of atoms that are mechanically coupled to one another, which gives rise to coordinated motions within the constraints of the folded structure. These motions are critical for many processes in molecular biology, including small-molecule and protein:protein binding interactions, catalytic cycles in enzymes, and allosteric communication between active sites and distal regulatory sites. Allostery in particular is now recognized to occur not only in classical oligomeric proteins like hemoglobin but also in monomers – and indeed may be inherent to nearly all protein structures (*Motlagh et al., 2014*; *Gunasekaran et al., 2004*). However, we do not yet understand at a fundamental level how mechanically coupled atoms underlie communication through protein structures, which prevents us from mapping their intrinsic allosteric 'circuitry'. Moreover, because protein

**eLife digest** Proteins perform many important jobs in each of the cells in our bodies, such as transporting other molecules and helping chemical reactions to occur. The part of the protein directly involved in these tasks is called the active site. Other areas of the protein can communicate with the active site to switch the protein on or off. This method of control is known as allostery.

Switching proteins on and off could help us to develop treatments for certain diseases. For example, a protein called PTP1B reduces how well cells can respond to insulin. Switching this protein off could therefore help to treat diabetes. However, much like it's hard to guess how a light switch is wired to a light bulb without seeing behind the walls, it is hard to predict which remote areas of a protein are 'wired' to the active site.

Keedy, Hill et al. have now used two complementary methods to examine the structure of PTP1B and find new allosteric sites. The first method captured a series of X-ray images from crystallized molecules of the protein held at different temperatures. This revealed areas of PTP1B that can move like windshield wipers to communicate with each other. The second method soaked PTP1B crystals in trays with hundreds of drug-sized molecules and assessed which sites on the protein the molecules bound to. The molecules generally bound to just a few sites of the protein. Further tests on one of these sites showed that it can communicate with the active site to turn the protein on or off.

Further work will be needed to develop drugs that could treat diabetes by binding to the newly identified allosteric sites in PTP1B. More generally, the methods developed by Keedy, Hill et al. could be used to study allostery in other important proteins.
DOI: https://doi.org/10.7554/eLife.36307.002

surfaces are large and intermolecular interactions are complex, it is difficult to predict which surface sites can bind an effector (such as a small molecule) that will allosterically communicate with the active site. These gaps severely limit our ability to elucidate natural allosteric regulatory mechanisms in biology, and to exploit allosteric circuitry in proteins for therapeutic intervention with perturbations such as small molecules.

One system that would benefit immensely from an improved mechanistic understanding of allostery is the archetypal protein tyrosine phosphatase, PTP1B (also known as PTPN1). PTP1B is highly validated as a therapeutic target for diabetes (*Elchebly et al., 1999*) and cancer (*Krishnan et al., 2014*) and has also been linked to Rett syndrome (*Krishnan et al., 2015*). Extensive efforts have been made to develop active-site inhibitors for PTP1B. Unfortunately, active-site inhibitors in general often bind non-specifically to homologous proteins, leading to off-target cellular effects (*DeDecker, 2000*). Moreover, the active sites of many enzymes, including phosphatases, are highly polar, and the polar inhibitors which bind to them often suffer from poor bioavailability (*Hardy and Wells, 2004*; *Zhang, 2001*). Attempts have been made to circumvent these limitations and selectively target the active site of PTP1B – for example, by linking non-hydrolyzable phosphotyrosine (pTyr) analogs that bind the active site with small-molecule fragments that bind in nearby, less conserved sites (*Zhang, 2017*). Nevertheless, no active-site inhibitors for PTP1B have reached clinical use, leading some to label PTP1B 'undruggable'.

By contrast, an allosteric inhibitor that binds to a less-conserved and less-polar surface site could bypass the limitations of active-site inhibitors. Two classes of compounds have been identified that allosterically inhibit PTP1B, although each has limitations. The first class of compounds are based on a benzbromarone (BB) scaffold and inhibit allosterically by binding to the space normally occupied by the regulatory C-terminal α7 helix (*Wiesmann et al., 2004*). Recent work combining mutagenesis, X-ray crystallography, NMR spectroscopy, and molecular dynamics simulations revealed how rotations of the α3 helix and a discrete switch of the catalytic WPD loop are impacted by these BB allosteric inhibitors (*Choy et al., 2017*). Unfortunately, the BB molecules were not successfully translated to the clinic. The second class are natural products, including a molecule called MSI-1436, that bind to multiple sites that primarily involve the disordered C-terminus (*Krishnan et al., 2014*). However, the binding poses were not structurally resolved, limiting our ability to understand the molecules' allosteric mechanism and rationally improve their potency. For example, a variant of MSI-1436 had

improved inhibition but a different response to mutations at the putative binding sites, suggesting an unknown change in mechanism (*Krishnan et al., 2018*). MSI-1436 passed Phase I clinical trials but was not advanced to Phase II (*Ghattas et al., 2016*). A new approach to revealing the intrinsic allosteric circuitry of proteins would reveal different opportunities to develop allosteric inhibitors for PTP1B that could potentially overcome the limitations of these existing molecules. Such an approach would additionally set the stage for efforts to dissect allosteric regulatory strategies in other biologically important phospho-signaling proteins.

Here, we have addressed the challenge of discovering unique opportunities for allosteric inhibition of PTP1B by taking advantage of two new techniques in X-ray crystallography that reveal minor conformational states of protein and ligands. First, multitemperature crystallography (*Keedy et al., 2015b*) can reveal previously hidden alternative conformations that enable biological functions. Here, we use this approach in PTP1B to reveal alternative conformations that are coupled to each other, forming an allosteric network. Our findings provide support for the previously hypothesized allosteric network in PTP1B that responds to BB inhibitors (*Choy et al., 2017*). Moreover, they reveal extensions of this network, including additional allosteric binding sites that are distinct from the BB site (*Figure 1*). Similar regions of PTP1B have been implicated as allosteric sites based on mutagenesis coupled with traditional cryogenic X-ray crystallography, molecular dynamics simulations, and NMR spectroscopy (*Choy et al., 2017*; *Cui et al., 2017*); here, we complement those studies by using multitemperature crystallography to reveal in atomic detail the conformational heterogeneity that allosterically links these sites to the active site. Second, high-throughput small-molecule fragment soaking and structure determination (*Collins et al., 2017*) has enabled new algorithms for revealing low-occupancy ligands (*Pearce et al., 2017*). We use this approach to comprehensively canvas the PTP1B surface with 1627 small-molecule fragments, 110 of which were structurally resolved in complex with PTP1B. The fragments cluster into 11 fragment-binding hotspots outside of the active site. To prioritize putative allosteric sites rather than benign binding sites, we focused on the subset of fragment-binding sites that were also conformationally coupled to the active site based on multitemperature crystallography of apo PTP1B. Strikingly, the sites chosen in this way

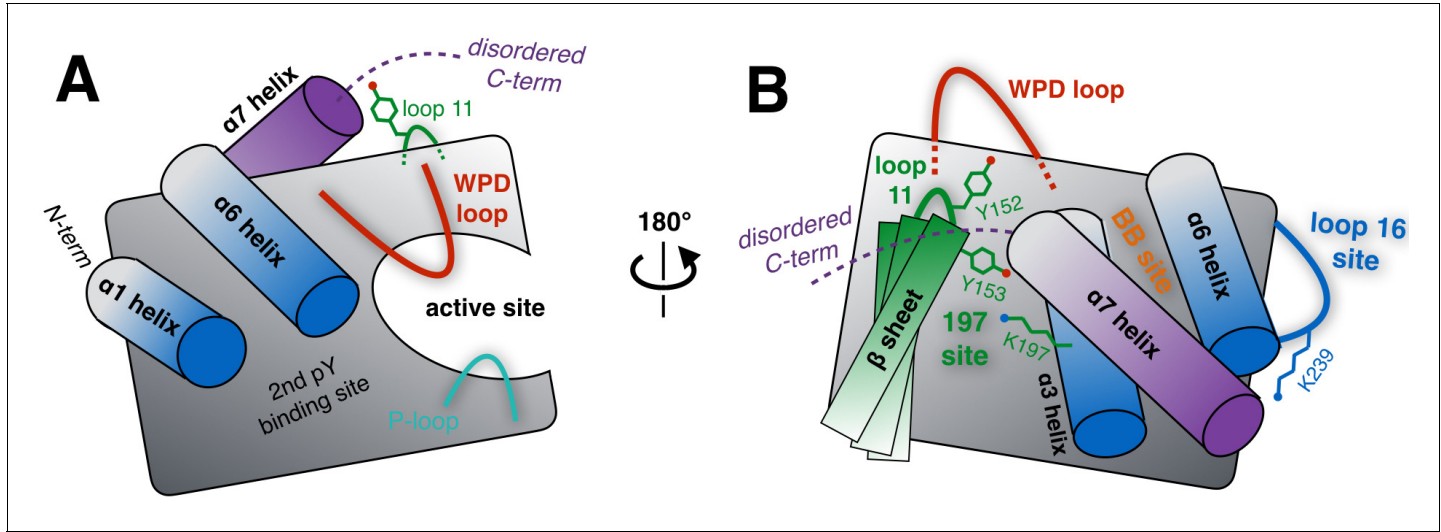

**Figure 1.** Schematic of key structural components in PTP1B. (**A**) The 'front side' of PTP1B features the active site covered by the dynamic catalytic WPD loop, as well as several other structural elements relevant to substrate recognition and binding. The α6 helix next to the WPD loop leads into the α7 helix and disordered C-terminus, which are positioned near loop 11 (partially occluded in this view). (**B**) On the 'back side' of PTP1B, with the view rotated by roughly 180°, the α7 helix and disordered C-terminus sit atop the α3 helix, the α6 helix, and the edge of the central β sheet including loop 11. The pocket between the α3 helix and the β sheet includes several sidechains which interact with each other, leading to the '197 site' (green). Elsewhere on the back side, a sidechain in loop 16 interacts with the α6-α7 connection to form the 'loop 16 site' or 'L16 site' (blue). These two allosteric sites are distinct from the previously established 'BB site' (orange) (*Wiesmann et al., 2004*), which is underneath the α7 helix that is displaced by BB allosteric inhibitor binding. As PTP1B transitions between its global states, many of the key structural components illustrated here undergo coordinated conformational changes, which together define the protein's intrinsic allosteric circuitry.
DOI: https://doi.org/10.7554/eLife.36307.003

bound more fragments than did any other sites – suggesting that conformational heterogeneity may be important for both allostery and ligand binding. Our work builds on previous studies of these sites in PTP1B (*Choy et al., 2017*; *Cui et al., 2017*), which did not report chemical matter that binds to them. Finally, we use covalently tethered small molecules (*Erlanson et al., 2004*) at one of these sites to confirm that it is functionally linked to enzyme activity, thereby supporting our predictions from multitemperature crystallography of the apo protein.

Overall, by highlighting promising allosteric sites and ligands that bind to them, our work may aid future development of potent non-covalent small-molecule allosteric inhibitors for PTP1B. More broadly, we illustrate a generalizable approach to characterizing and exploiting coupled conformational heterogeneity to enable long-range control of protein function.

## Results

### Identifying allosterically coupled residues with multitemperature crystallography

To identify allosteric sites in PTP1B that can communicate with the active site, we searched for regions of the protein whose conformational heterogeneity is coupled to that of the active site. We began by examining the conformational heterogeneity of the active-site WPD loop. Transition of this loop from the open to the closed state is rate-limiting for catalysis (*Whittier et al., 2013*). In the only available apo crystal structure of PTP1B in which the WPD loop is free from crystal-lattice contacts (PDB ID 1sug) (*Pedersen et al., 2004*), the loop is modeled in the closed state. However, low-contour electron density can reveal hidden alternative conformations in protein crystal structures (*Lang et al., 2010*; *Fraser et al., 2011*). We therefore investigated the electron density near the WPD loop in the apo structure more closely (*Figure 2B*).

Surprisingly, upon closer inspection, the electron density strongly suggests a significant population for the open state as well (*Figure 2C*, left). Our re-refined model with both open and closed states as alternative conformations visually accounts for the electron density around this loop much better than the original model (*Figure 2C*, left). By contrast, when we re-refined 36 other available crystal structures of PTP1B complexed with active-site inhibitors using both open and closed loop states as putative alternative conformations, Fo-Fc difference electron density and the bimodal distribution of refined occupancies indicated the single-state models were a better fit (*Figure 2—figure supplement 1*). These results suggest that, even in the crystal, apo PTP1B samples both WPD loop states and that active-site inhibitors then lock the loop either fully open or fully closed.

To better characterize the conformational heterogeneity of the WPD loop in apo PTP1B, we collected X-ray datasets at several elevated temperatures including 180 K, 240 K, and 278 K ('room temperature') in addition to the 100 K ('cryogenic') model from the PDB, all at better than 2 Å resolution (*Table 1*). Each complete dataset was obtained from a single crystal, and crystallographic statistics indicated that radiation damage was not a concern even at the elevated temperatures (*Diederichs, 2006*) (*Figure 2—figure supplement 3*). We built an initial multiconformer model for each temperature using the automated algorithm qFit (*Keedy et al., 2015a*). These models are parsimonious in that each atom has alternative positions only if justified by the experimental data, and a single position otherwise. Such models are equally good and usually better explanations of the experimental X-ray data (*Keedy et al., 2015a*; *van den Bedem et al., 2009*), and have been used to understand many biologically relevant phenomena at protein:water interfaces (*Keedy et al., 2014*), dynamic enzyme active sites (*Keedy et al., 2015b*; *Fraser et al., 2009*), and allosteric networks perturbed by mutations (*van den Bedem et al., 2013*). We then manually refined alternative conformations for protein, buffer components, and solvent. In particular, we took advantage of the wealth of available structures of PTP1B in the PDB (*Berman et al., 2000*) to sample coordinates for putative alternative conformations; in many cases, these conformations explained missing regions with positive Fo-Fc electron density that would have otherwise been difficult to model. Removing the alternative conformations and re-refining the resulting single-conformer models, either with or without automated solvent placement, yields deteriorated statistics (*Table 1— source data 1*), which confirms that the multiconformer models are appropriate explanations of the experimental data at each temperature.

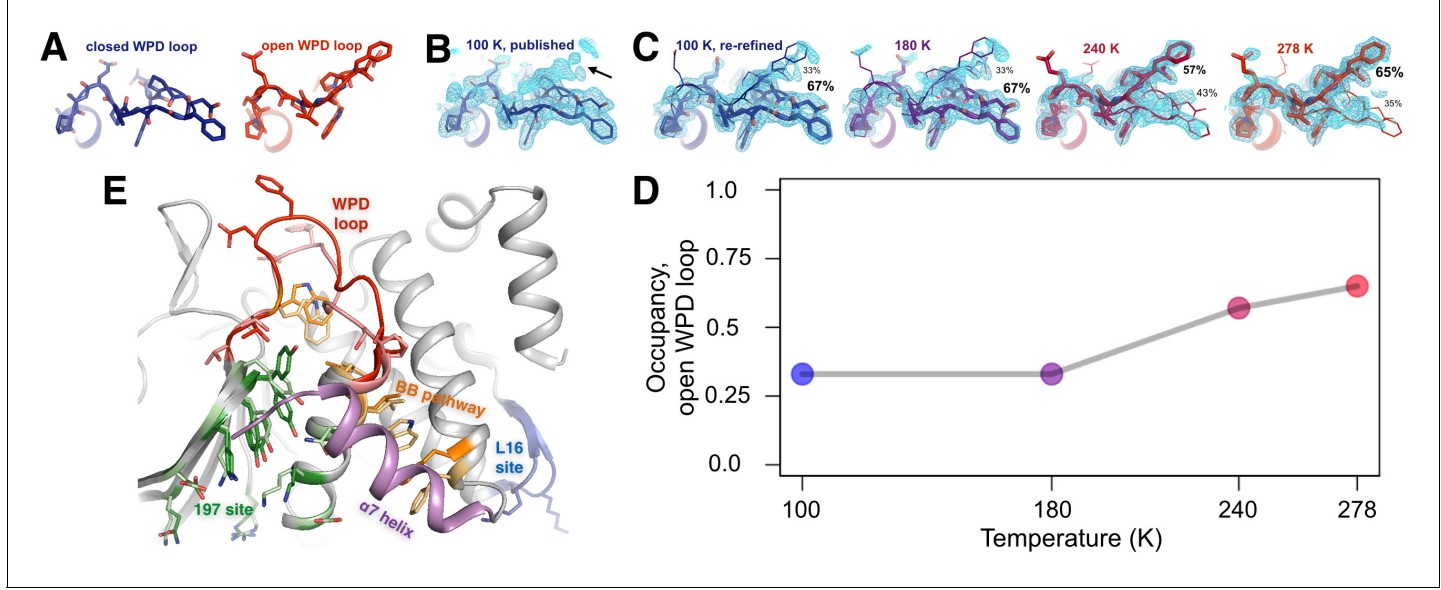

**Figure 2.** The conformational ensemble of the active-site WPD loop and allosterically coupled regions. (**A**) The active-site WPD loop in PTP1B adopts either a closed conformation (example from PDB ID 1sug) or an open conformation (example from PDB ID 1t49). View from the 'front side' of PTP1B. (**B**) In the previously published apo structure of PTP1B, solved at 100 K (PDB ID 1sug), 0.8 σ 2Fo-Fc electron density (cyan) supports the modeled closed conformation, but substantial electron density remains unexplained (arrow). (**C**) Adding the open conformation of the WPD loop as a secondary conformation at partial occupancy accounts for this electron density. In structures solved at different elevated temperatures, electron density for the open conformation becomes more prominent as its occupancy (labeled) relative to the closed conformation increases. (**D**) The occupancy of the open conformation increases non-linearly with temperature. (**E**) Overall roadmap of allostery on the 'back side' of PTP1B, with the allosteric 197 site and loop 16 (L16) site highlighted in the context of the larger allosteric network including the previously established BB site, a7 helix, and WPD loop. Sidechains are shown in stick representation for several key residues in the WPD loop and allosteric regions. For those residues with alternative conformations at 278 K, both open-state (darker hues) and closed-state (lighter hues) conformations are shown. The viewing orientation in (**A–C**) is as in *Figure 1A* ('front side' of PTP1B), except zoomed in on the active site (labeled in *Figure 1A*). The viewing orientation in (**E**) is as in *Figure 1B* ('back side' of PTP1B).
DOI: https://doi.org/10.7554/eLife.36307.004

The following video and figure supplements are available for figure 2:

**Figure supplement 1.** The WPD loop adopts multiple conformations only in the absence of inhibitors.
DOI: https://doi.org/10.7554/eLife.36307.005

**Figure supplement 2.** Mutually exclusive partial-occupancy protein and solvent atoms complicate model building.
DOI: https://doi.org/10.7554/eLife.36307.006

**Figure supplement 3.** Radiation damage is minimal across all datasets.
DOI: https://doi.org/10.7554/eLife.36307.007

**Figure 2—video 1.** Movie version with five scenes: *Figure 2B* then *Figure 2C*.
DOI: https://doi.org/10.7554/eLife.36307.008

**Figure 2—video 2.** Movie version of *Figure 2E*.
DOI: https://doi.org/10.7554/eLife.36307.009

The WPD loop adopts both the open and closed conformations across this range (*Figure 2C*) and the population of the open vs. closed states was sensitive to temperature (*Figure 2D*). The loop is approximately 67% closed at 100 K, but 65% open at 278 K. These occupancies evolve non-linearly (*Keedy et al., 2015b*) at intermediate temperatures.

Overall, we also observed temperature-dependent conformational heterogeneity for several other regions of PTP1B, including the previously characterized BB allosteric site, plus additional sites we refer to as the '197 site' and the 'loop 16 (L16) site'. These regions are all contiguous in the structure (*Figure 2E*), suggesting that they together constitute an expanded collective allosteric network in PTP1B that is coupled to the WPD loop. The manner in which they are connected is described in detail in the following sections.

**Table 1.** Crystallographic statistics for multitemperature and mutant X-ray datasets.

Overall statistics given first (statistics for highest-resolution bin in parentheses). For WT apo, 100 K: statistics are taken from our remodeled structure where appropriate, or from the original PDB ID 1sug when possible otherwise, or given as '—" where unavailable.

| | WT apo, 100 K | WT* apo, 180 K | WT* apo, 240 K | WT* apo, 278 K | WT* with BB3, 278 K | K197C apo, 100 K | K197C tethered to 2, 100 K |
|---|---|---|---|---|---|---|---|
| PDB ID | 6B90 | 6B8E | 6B8T | 6B8X | 6B8Z | 6BAI | 6B95 |
| Number of crystals used | 1 | 1 | 1 | 1 | 1 | 1 | 1 |
| Wavelength (Å) | 0.8110 | 1.115869 | 1.115869 | 0.9795 | 1.11583 | 1.11583 | 1.11583 |
| Resolution range (Å) | 33.60–1.95 (2.02–1.95) | 19.18–1.82 (1.89–1.82) | 62.54–1.85 (1.92–1.85) | 31.31–1.74 (1.80–1.74) | 43.88–1.8 (1.86–1.8) | 75.57–1.95 (2.02–1.95) | 43.98–1.95 (2.02–1.95) |
| Space group | P 31 2 1 | P 31 2 1 | P 31 2 1 | P 31 2 1 | P 31 2 1 | P 31 2 1 | P 31 2 1 |
| Unit cell (Å, °) | 88.12 88.12 103.90 90 90 120 | 88.57 88.57 104.32 90 90 120 | 89.44 89.44 106.00 90 90 120 | 89.52 89.52 106.25 90 90 120 | 89.65 89.65 106.39 90 90 120 | 87.27 87.27 104.10 90 90 120 | 87.96 87.96 104.63 90 90 120 |
| Total reflections | — | 339461 (33574) | 256701 (25484) | 299041 (30678) | 771858 (76158) | 653798 (63759) | 170094 (16845) |
| Unique reflections | 34486 (3371) | 42866 (4223) | 42343 (4156) | 50486 (4960) | 46311 (4561) | 33409 (3296) | 34596 (3387) |
| Multiplicity | 7.4 (—) | 7.9 (7.9) | 6.1 (6.1) | 5.9 (6.2) | 16.7 (16.7) | 19.6 (19.3) | 4.9 (5.0) |
| Completeness (%) | 99.9 (100.0) | 99.9 (100.0) | 100.0 (100.0) | 99.0 (98.3) | 100.0 (100.0) | 98.06 (98.51) | 99.81 (99.88) |
| Mean I/sigma(I) | 26.4 (4.4) | 21.40 (1.46) | 15.89 (1.46) | 17.15 (1.49) | 11.69 (0.75) | 25.06 (0.94) | 12.04 (0.81) |
| Wilson B (Å$^2$) | 24.12 | 23.69 | 28.48 | 19.48 | 34.96 | 46.3 | 38.82 |
| R$_{merge}$ | 0.073 (0.515) | 0.089 (1.483) | 0.079 (1.215) | 0.098 (1.209) | 0.148 (2.664) | 0.076 (3.056) | 0.088 (1.852) |
| R$_{meas}$ | — | 0.096 (1.585) | 0.086 (1.329) | 0.108 (1.318) | 0.153 (2.748) | 0.078 (3.138) | 0.098 (2.071) |
| R$_{pim}$ | — | 0.034 (0.556) | 0.035 (0.535) | 0.044 (0.520) | 0.038 (0.672) | 0.018 (0.705) | 0.044 (0.916) |
| CC1/2 | — | 0.999 (0.520) | 0.999 (0.560) | 0.999 (0.496) | 0.997 (0.366) | 1.000 (0.438) | 0.999 (0.292) |
| CC* | — | 1.000 (0.827) | 1.000 (0.847) | 1.000 (0.814) | 0.999 (0.732) | 1.000 (0.781) | 1.000 (0.672) |
| Reflections used in refinement | 34486 (3371) | 42862 (4223) | 42343 (4156) | 50486 (4959) | 46309 (4561) | 33302 (3296) | 34576 (3386) |
| Reflections used for R$_{free}$ | 1356 (128) | 1689 (165) | 1669 (163) | 1993 (196) | 1820 (176) | 1310 (135) | 1362 (135) |
| R$_{work}$ | 0.1580 (0.1881) | 0.1708 (0.2781) | 0.1674 (0.2760) | 0.1752 (0.3511) | 0.1675 (0.3116) | 0.2061 (0.3365) | 0.1858 (0.3295) |
| R$_{free}$ | 0.1926 (0.2090) | 0.1997 (0.2911) | 0.2123 (0.2869) | 0.2059 (0.3888) | 0.1978 (0.3017) | 0.2569 (0.3292) | 0.2307 (0.3686) |
| CC$_{work}$ | — | 0.963 (0.753) | 0.958 (0.789) | 0.964 (0.673) | 0.964 (0.709) | 0.934 (0.618) | 0.957 (0.574) |
| CC$_{free}$ | — | 0.948 (0.744) | 0.908 (0.677) | 0.948 (0.739) | 0.961 (0.757) | 0.994 (0.484) | 0.938 (0.294) |
| Non-H atoms | 2997 | 3021 | 3149 | 3260 | 2709 | 2502 | 2541 |
| Macromolecule atoms | 2657 | 2687 | 2875 | 3011 | 2460 | 2377 | 2370 |
| Ligand atoms | 46 | 40 | 34 | 24 | 82 | 16 | 36 |
| Solvent atoms | 294 | 294 | 240 | 225 | 167 | 109 | 135 |
| Protein residues | 298 | 298 | 299 | 298 | 289 | 282 | 285 |
| RMS bonds (Å) | 0.010 | 0.011 | 0.011 | 0.012 | 0.011 | 0.015 | 0.013 |
| RMS angles (°) | 1.01 | 1.02 | 1.10 | 1.08 | 1.14 | 1.30 | 1.15 |
| Ramachandran favored (%) | 97.64 | 97.64 | 97.98 | 95.95 | 97.19 | 95.00 | 95.76 |
| Ramachandran allowed (%) | 2.03 | 2.03 | 1.68 | 3.72 | 2.11 | 4.64 | 2.83 |
| Ramachandran outliers (%) | 0.34 | 0.34 | 0.34 | 0.34 | 0.70 | 0.36 | 1.41 |

*Table 1 continued on next page*

*Table 1 continued*

| | WT apo, 100 K | WT* apo, 180 K | WT* apo, 240 K | WT* apo, 278 K | WT* with BB3, 278 K | K197C apo, 100 K | K197C tethered to 2, 100 K |
|---|---|---|---|---|---|---|---|
| Rotamer outliers (%) | 1.35 | 2.67 | 1.87 | 2.38 | 2.19 | 6.44 | 4.91 |
| Clashscore | 2.79 | 3.31 | 3.62 | 2.98 | 2.79 | 6.06 | 3.36 |
| MolProbity score | 1.24 | 1.52 | 1.36 | 1.65 | 1.47 | 2.29 | 1.95 |
| Average B (Å²) | 29.35 | 30.22 | 36.50 | 28.06 | 43.58 | 55.81 | 49.66 |
| Average B, macromolecule (Å²) | 28.14 | 29.11 | 35.51 | 27.01 | 43.30 | 56.02 | 49.53 |
| Average B, ligands (Å²) | 47.43 | 48.08 | 60.29 | 67.33 | 41.13 | 45.41 | 53.91 |
| Average B, solvent (Å²) | 37.46 | 37.90 | 44.96 | 38.01 | 48.96 | 52.76 | 50.82 |

DOI: https://doi.org/10.7554/eLife.36307.010

The following source data is available for Table 1:

Source data 1. Multiconformer models best explain PTP1B X-ray data across temperatures.

R-factors are reported for the deposited multiconformer models for multitemperature datasets in *Table 1* vs. single-conformer models derived by removing alternative conformations and re-refining with Phenix either with default parameters ('without solvent picking') or with automated water placement turned on by adding the flag 'ordered_solvent = True' ('with solvent picking') for 12 macro-cycles. For the new single-conformer models, R-factors are given both for state A with state B's alternative conformations removed, and for state B with state A's alternative conformations removed, to confirm that either option for a single-conformer model is worse than a multiconformer model. Overall, regardless of the choice of solvent parameters, at each temperature the multiconformer model has lower (better) $R_{work}$ and $R_{free}$.

DOI: https://doi.org/10.7554/eLife.36307.011

## Multitemperature crystallography of the BB allosteric site

To connect these multitemperature structures to known allosteric regulatory mechanisms, we first turned to a benzbromarone derivative compound (here referred to as BB2) that binds to an allosteric site >12 Å away from the active site and inhibits enzyme activity (*Wiesmann et al., 2004*). The authors of the study reporting BB2 described a series of induced conformational changes that begins with BB2 directly displacing Trp291 to disorder the entire C-terminal α7 helix, and ends with Phe191 χ2 dihedral-angle rotations clashing with the WPD loop anchor residue Trp179 to stabilize the open state. We tested the hypothesis that these allosterically inhibited conformations pre-exist in apo PTP1B by examining these regions in our multitemperature apo crystal structures. Indeed, in apo PTP1B the α7 helix is more ordered at lower temperatures but more disordered at higher temperatures (*Figure 3A*). Also, Trp179 and Phe191 adopt dual conformations at higher temperatures (*Figure 3B*) that coincide well with the apo and allosterically inhibited conformations (*Figure 3C*). We also see alternative conformations at high temperatures for several residues within and directly flanking the WPD loop (Arg221, Pro185, Trp179, Phe269) which have been implicated as being important for a CH/π switch during WPD loop opening/closing (*Choy et al., 2017*) (*Figure 3—figure supplement 1*). Multiple conformations for Leu192 were more difficult to detect at higher temperatures in apo PTP1B. This is likely because Leu192 shifts more subtly between the 100 K apo and allosterically inhibited conformations, which is also consistent with a recent report that Leu192 is a relatively static inter-helical 'wedge' (*Choy et al., 2017*). Taken together, these results suggest that BB2 stabilizes a subset of pre-existing conformations in apo PTP1B.

We additionally solved a high-resolution (1.80 Å, *Table 1*) structure of PTP1B in complex with BB3 (which differs from BB2 only by an extra terminal aminothiazole group) at 273 K and found it to be very similar to the 100 K structures with BB3 (PDB ID 1t4j) and with BB2 (PDB ID 1t49) despite the difference in temperature (*Figure 3—figure supplement 2*). However, two interesting features are evident at 273 K. First, at 273 K but not at 100 K, modeling BB3 with a single conformer leads to Fo-Fc difference electron density peaks at both ends of the molecule (*Figure 3—figure supplement 3A*). To account for these peaks in the map, it is necessary to add a second alternative conformer to the model, which includes a translation at one end and dihedral-angle changes at the other end

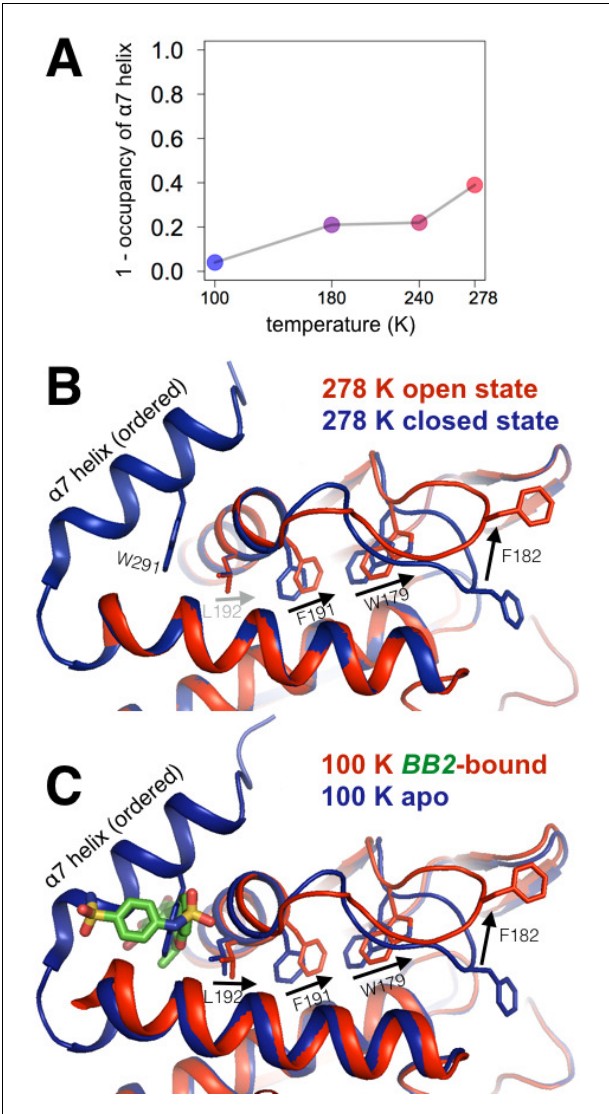

**Figure 3.** Multitemperature crystallography of apo PTP1B recapitulates an allosteric mechanism. (**A**) In apo PTP1B, the occupancy of the α7 helix decreases (i.e. the helix becomes more disordered) with temperature. The helix was modeled with one conformation and its occupancy was refined; the remaining occupancy corresponds to the disordered state of the helix. (**B**) Several residues that allosterically link α7 and the active-site WPD loop also undergo shifts with temperature. (**C**) These additional conformations match the state trapped by the allosteric inhibitor BB2 (PDB 1t49) (*Wiesmann et al., 2004*) which binds >12 Å away from the active site. The viewing orientation in (**B–C**) is as in *Figure 1A* ('front side' of PTP1B), except slightly zoomed in.

DOI: https://doi.org/10.7554/eLife.36307.012

The following figure supplements are available for figure 3:

**Figure supplement 1.** Alternative conformations in apo PTP1B recapitulate a reported conformational switching mechanism in the active site during WPD loop closing.

DOI: https://doi.org/10.7554/eLife.36307.013

**Figure supplement 2.** Allosteric inhibitor binding quenches conformational heterogeneity regardless of temperature.

DOI: https://doi.org/10.7554/eLife.36307.014

**Figure supplement 3.** Allosteric-inhibitor-bound PTP1B has some low-occupancy conformations only at 273 K.

DOI: https://doi.org/10.7554/eLife.36307.015

(*Figure 3—figure supplement 3B*). Chemical changes to BB3 designed to eliminate this remaining heterogeneity could potentially improve affinity and inhibition.

Second, at 273 K, we observe significant electron density just above BB3 (*Figure 3—figure supplement 3C*). Modeling a reordered, non-helical conformation of α7 explains this density well, and places Trp291 in good position for aromatic stacking interactions with BB3 and other interactions with nearby sidechains on the α3 helix (*Figure 3—figure supplement 3D*). Trp291 is displaced by BB3 or BB2 binding in a striking example of molecular mimicry (*Wiesmann et al., 2004*) (*Figure 3C*). Our 273 K data suggest that a subsequent reordering of the α7 polypeptide occurs, which may contribute to the affinity of BB3 for PTP1B. In contrast to our 273 K data, electron density in this region is weak in the 100 K structures with BB3 and BB2. However, in the 100 K structure with BB1, a different derivative of the BB scaffold, α7 also reorders – but adopts a significantly different conformation than we observe at 273 K with BB3 (*Figure 3—figure supplement 3E,G*). Together, these results suggest that in addition to being a major allosteric hub when ordered (*Choy et al., 2017*), α7 is also quite malleable when disordered, and may interact in diverse ways with bound ligands – behavior which is similar to the mechanism proposed for inhibitors that bind via the disordered C-terminus beyond α7 (*Krishnan et al., 2014*).

## Multitemperature crystallography of the allosteric loop 16 site

We also observed temperature-dependent ordering in a loop (loop 16, L16; residues 237–243) that sits underneath the α6-α7 junction just beyond the BB binding site. By contrast to lower temperature (*Figure 4A*), the electron density for L16 at higher temperature (*Figure 4B*) clearly reveals an alternative conformation with its backbone shifted by >5 Å from the primary conformation (*Figure 4D*). Modeling this alternative loop conformation back into the lower-temperature models and refining its occupancy reveals a temperature dependence (*Figure 4E*, *Figure 4—figure supplement 1*) that is qualitatively similar to the temperature dependences of the WPD loop. Remarkably, this L16 alternative conformation sampled by apo PTP1B matches the L16 conformation when PTP1B is allosterically inhibited by BB2 (*Figure 4C*). This rearrangement provides further evidence that BB2 selects pre-existing, globally dispersed conformations rather than inducing new ones.

The L16 site is seemingly coupled to the α6 helix: Lys239 from L16 H-bonds with Ile281 from α6 in the global closed state, but not in the global open state in which L16 adopts its alternative conformation. Because α6 is directly coupled to the α7 order-disorder transition, we therefore propose that the L16 site is a component of the collective allosteric network in PTP1B.

The L16 site was not identified as part of the allosteric network in PTP1B based on a study using mutagenesis, NMR, and MD (*Choy et al., 2017*). However, in a more recent study, several residues lining what we call the L16 site (including Met3, Lys237, and Ser242) were included in a region called 'Cluster II', which was suggested to be a previously unidentified allosteric site based on reciprocal NMR chemical shift perturbations upon mutation of this site or the WPD loop (*Cui et al., 2017*). Our work here using multitemperature crystallography complements these findings by independently identifying this allosteric site using a new methodology, and by revealing in atomic detail how multiple conformational states at the L16 site may aid communication with the active site. Interestingly, a separate approach combining molecular dynamics and machine learning also recently pointed to this area as a potential 'cryptic' binding site (*Cimermancic et al., 2016a*). Therefore, the L16 site may be not only energetically coupled to the active site, but also capable of forming an underappreciated small-molecule binding pocket via the conformational heterogeneity we observe.

## Multitemperature crystallography of the allosteric 197 site

In addition to the temperature-dependent conformational heterogeneity observed at the BB site and L16 site, we observed residues with temperature-sensitive conformational heterogeneity in the '197 site' (*Figure 5*). Moreover, the alternative conformations of several residues in this region have a pattern of steric incompatibility with multiple states of the WPD loop and α7 helix, suggesting that the 197 site may be mechanistically linked to the active site in a similar way as the BB binding site.

A major link between the WPD loop and the 197 site is Tyr152. When the WPD loop is closed and the α7 helix is ordered, Tyr152 adopts a 'down rotamer' (*Figure 5—figure supplement 1*, red). By contrast, when the WPD loop is open and the α7 helix is disordered, the 278 K electron density suggests that Tyr152 adopts an 'up rotamer' (*Figure 5—figure supplement 1C*, orange). However,

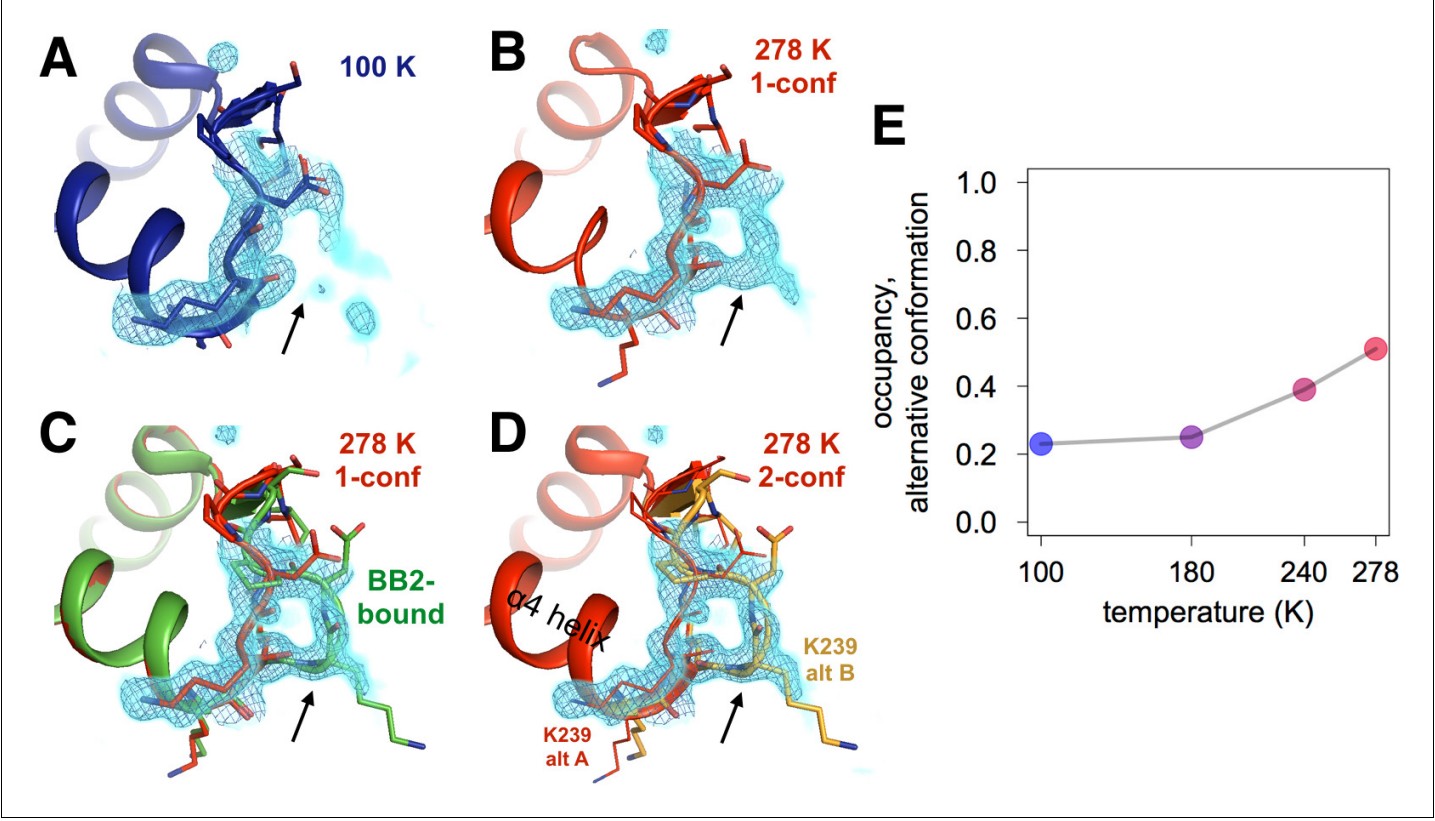

**Figure 4.** Both an allosteric inhibitor and high temperature favor an alternative conformation for an α7-coupled loop 16. (**A**) At low temperature, loop 16 (residues 237–243, bottom right) is single-conformer, as evidenced by 2Fo-Fc electron density contoured at 1.0 σ (cyan volume) and at 1.0 σ (blue mesh). (**B**) At high temperature, when the protein is modeled as single-conformer, the electron density suggests the existence of an alternative conformation. (**C**) The structure with BB2 bound (>12 Å away) (PDB ID 1t49) perfectly explains the mysterious electron density. (**D**) The final 278 K dual-conformation model is a good explanation of the data. (**E**) The refined occupancy of the alternative conformation (state 'B') in apo PTP1B increases continuously but non-linearly with temperature. The viewing orientation in (**A–D**) is as in *Figure 1B* ('back side' of PTP1B), except zoomed in on the loop 16 site (labeled in *Figure 1B*).

DOI: https://doi.org/10.7554/eLife.36307.016

The following figure supplement is available for figure 4:

**Figure supplement 1.** The conformational distribution of the α7-coupled loop 16 titrates with temperature.

DOI: https://doi.org/10.7554/eLife.36307.017

difference electron density peaks remain (*Figure 5—figure supplement 1C*) that indicate the presence of the down rotamer as an alternative conformation. Consistent with this interpretation, modeling just the additional down rotamer is insufficient to explain the density (*Figure 5—figure supplement 1D*). These two rotamers are accommodated in the WPD-loop-open state by a shift of the L11 backbone (*Figure 5—figure supplement 1*). The down rotamer is sterically incompatible with phosphorylation of Tyr152, which occurs in vivo (*Bandyopadhyay et al., 1997*; *Rhee et al., 2001*), suggesting that the up rotamer may have additional regulatory roles. Tyr152 in the L11 backbone conformation with just the down rotamer (red in *Figure 5—figure supplement 1*) is sterically incompatible with the open WPD loop conformation (*Figure 5—figure supplement 1E*). Similarly, the Tyr152 up rotamer is sterically incompatible with the ordered α7 conformation (*Figure 5—figure supplement 1E*). In turn, α7 is conformationally synchronized with the WPD loop (*Figure 3A* and *Figure 2D*) and is a key hub connecting loop 11 and the WPD loop (*Choy et al., 2017*). These results together suggest that the allosteric circuitry of PTP1B involving Tyr152 is complex and multibody. Tyr152 likely exemplifies a population shuffling mechanism whereby mixtures of microstates (rotameric state of Tyr152) exchange on a fast timescale as the protein transitions between macrostates (WPD loop state, α7 ordering, and L11 backbone shifting) on a slower timescale (*Smith et al., 2015*). Our findings thus shed additional light on the mechanism by which loop 11 allosterically

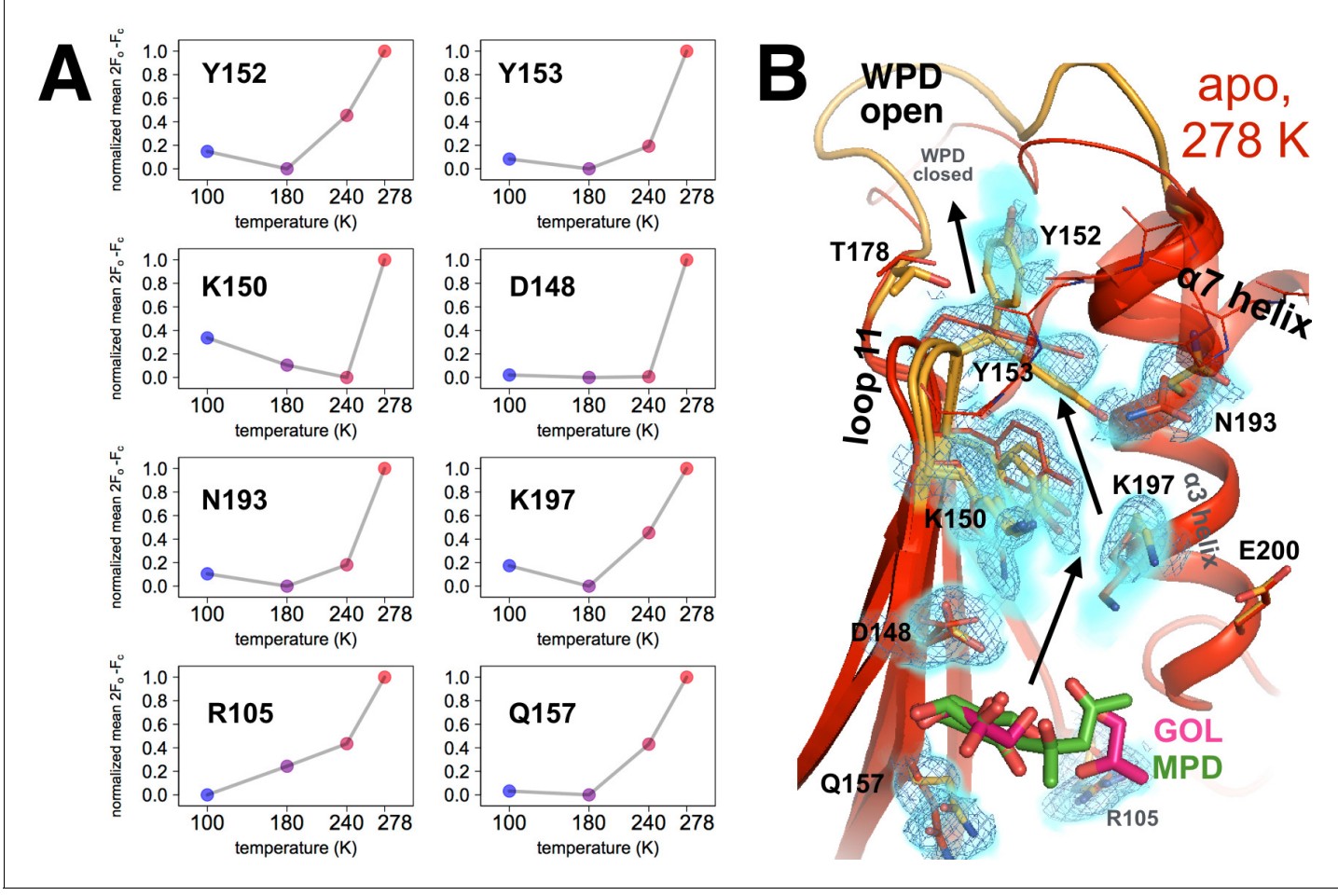

**Figure 5.** Coupled conformational heterogeneity leads to the allosteric 197 site. (A) Several residues distinct from both the active site and a previously characterized allosteric network each have minor alternative conformations that become more populated with temperature. This is quantified by the sum of 2Fo-Fc electron density values for the centers of atoms that are unique to the minor state (defined as being at least 1.0 Å away from any atoms in the major state), normalized across temperatures from 0 to 1 for each residue. (B) These residues colocalize to a region of the protein surrounded by loop 11 (top-left), the quasi-ordered α7 helix (top-right), and the α3 helix (right), including the eponymous K197. 2Fo-Fc electron density contoured at 0.6 σ (cyan volume) and at 0.8 σ (blue mesh) justify multiple conformations for these residues in our 278 K apo model, as quantified in (A). The alternative conformations of these residues appear to interact with one another and thus may be allosterically coupled. Ordered crystallization mother liquor or cryoprotectant molecules (glycerols in pink, from the PDB and our structures, or MPD molecules in green, from the PDB) can be present at the terminus of this allosteric pathway, suggesting it may be amenable to binding other small molecules. The viewing orientation in B) is as in *Figure 1B* ('back side' of PTP1B), except zoomed in on the 197 site (labeled in *Figure 1B*).

DOI: https://doi.org/10.7554/eLife.36307.018

The following video and figure supplements are available for figure 5:

**Figure supplement 1.** Alternative conformations in apo PTP1B recapitulate and expand upon reported coupling between loop 11 and α3.
DOI: https://doi.org/10.7554/eLife.36307.019

**Figure supplement 2.** The allosteric 197 site has local sequence differences in related PTPs.
DOI: https://doi.org/10.7554/eLife.36307.020

**Figure supplement 3.** Mutations along the 197 site's allosteric pathway reduce enzyme activity.
DOI: https://doi.org/10.7554/eLife.36307.021

**Figure supplement 4.** Flexible aromatic residues complete an allosteric circuit.
DOI: https://doi.org/10.7554/eLife.36307.022

**Figure 5—video 1.** Movie version of *Figure 5B*.
DOI: https://doi.org/10.7554/eLife.36307.023

communicates with the active site, thus complementing other recent studies using mutagenesis, MD, and NMR to map allostery in PTP1B (*Choy et al., 2017*; *Cui et al., 2017*).

In our datasets at temperatures above 100 K, the electron density suggests a complex interplay between alternative conformations for Asn193 on the α3 helix and Tyr152 on loop 11 (L11) (*Figure 5—figure supplement 1*). Asn193 is part of the α3 helix (residues 187–202), which immediately follows the WPD loop in sequence. The N-terminal region of this helix (through Phe196) rotates by 2–20°, resulting in shifts of 0.2–0.7 Å for some Cα atoms, based on 100 K crystal structures of apo (WPD-open) vs. active-site-inhibitor-bound (WPD-closed) PTP1B (*Choy et al., 2017*). Similarly, the multiconformer model for our 278 K apo dataset includes alternative backbone conformations for the WPD loop and the beginning of α3, through Asn193 plus Phe196-Lys197 (this is a conservative interpretation of which residues in the helix have alternative backbone conformations). Our results suggest that α3 inherently shifts as the protein transitions between its global macrostates, even in the apo state.

Strikingly, several residues propagating down L11 from Tyr152, and down α3 from Asn193, also adopt multiple conformations at higher temperatures (*Figure 5*). These residues colocalize in a shallow pocket nestled between loop 11, the β4 and β5 strands, and the α3 and α7 helices. We refer to this area here as the '197 site' because the sidechain of Lys197 extends into the pocket. Our analysis indicates a complex, interconnected network involving multiple aromatic stacking, hydrogen-bonding, van der Waals, and electrostatic interactions. To complement this model-based assessment with a map-based approach, for several residues in the pocket we quantified electron density as a function of temperature for atom positions that are unique to the minor conformation (i.e. do not overlap with any atoms in the major conformation), reasoning that residues which respond to temperature similarly may be conformationally coupled (*Keedy et al., 2015b*). The population of each minor conformation increases non-linearly with temperature (*Figure 5A*) in a similar fashion as the open state of the WPD loop (*Figure 2D*) and the disordered state of the α7 helix (*Figure 3A*), in support of the idea that these various regions of the protein are mutually conformationally coupled.

We next discuss several similarities and a few differences between what we refer to as the 197 site and similar regions implicated by other recent studies of allostery in PTP1B. First, in addition to predicting the L16 site (see above), reciprocal chemical shift changes upon mutation suggested that several residues at both ends of the 197 site (Tyr152, Tyr153, Lys150, Arg105) are part of a region referred to as 'Cluster I' that is allosterically linked to the active site (*Cui et al., 2017*). However, that study did not implicate additional key residues on the α3 helix, for example Asn193 and Lys197. Second, mutagenesis, NMR, and cryogenic crystallography implicated several elements of our 197 site as being part of the larger allosteric network in PTP1B: loop 11 (including Tyr152 and Tyr153), the α3 helix (especially Asn193), and the α7 helix (*Choy et al., 2017*). Chemical-shift-restrained molecular dynamics simulations further suggested that Tyr152 on loop 11 and Asn193 on the α3 helix have mutually coupled alternative conformations (*Choy et al., 2017*). However, here we highlight additional residues (e.g. Asp148 and Glu157 on the β strands on either end of loop 11) as being conformationally coupled to each other and to the rest of the allosteric network and the active site, and which may collectively form a binding pocket. Therefore, our work accomplishes two things with regard to these existing studies. First, we add support to their findings by reaching similar conclusions using orthogonal methods. Second, we complement the other studies by revealing additional amino acid residues that may play roles in binding and allosteric communication at the 197 site.

We also emphasize that the 197 site is structurally distinct from the two allosteric sites that have previously been targeted with small molecules to achieve inhibition (*Krishnan et al., 2014*; *Wiesmann et al., 2004*), so any small molecules that bind to the 197 site would represent a distinct strategy for inhibiting PTP1B. Surprisingly, in all of our multitemperature apo structures, ordered glycerols are present not only in the active site as mentioned above but also in the 197 site (*Figure 5—figure supplement 2*), and MPD also binds here in another published structure (PDB ID 2cm2). These observations suggest that the 197 site may be bindable by other small molecules. We therefore hypothesized that binding of a small molecule to the 197 site could propagate changes in conformational heterogeneity to the WPD loop to interfere with catalysis.

To test whether more directed perturbations to the 197 site can allosterically modulate enzyme function, we introduced 'dynamically destructive' mutations (Y152G, Y153A, K197A) that were predicted to preserve the protein's general structure, yet interfere with the conformational heterogeneity along the putative allosteric pathway lining the 197 site by removing interactions between

alternative conformations. For Y152 we chose a mutation to glycine instead of alanine to more fully disengage residue 152 from the WPD loop, given that the Cβ and Hβ atoms of Y152 sterically engage with the WPD loop (*Figure 5—figure supplement 1E*). All three mutations indeed reduce catalytic efficiency, to varying extents: the mutation nearest to the WPD loop (Y152G) reduced $k_{cat}/K_M$ the most, and the mutation farthest from the WPD loop (K197A) reduced $k_{cat}/K_M$ the least (*Figure 5—figure supplement 3*). Our results are generally in line with reported effects for the Y152A + Y153A ('YAYA') double mutation (*Choy et al., 2017*) and for the Y153A single mutation (*Cui et al., 2017*); small discrepancies may be due in part to differences in the length of the protein construct being used. Overall, our results illustrate that local perturbations in the vicinity of the allosteric 197 site can impact catalysis.

Overall, we describe a large, collectively coupled allosteric network on one contiguous face of the protein (*Figure 2E*). This network is interconnected not only on the surface, but also within the hydrophobic core. For example, Tyr176 adopts alternative sidechain conformations at higher temperatures that differ by a small rotation of the relatively non-rotameric $\chi 2$ dihedral angle (*Lovell et al., 2000*) (*Figure 5—figure supplement 4*). The two conformations of Tyr176 are structurally compatible with different conformations of the surface-exposed Tyr152 (*Choy et al., 2017*; *Cui et al., 2017*) in one direction, and of the buried Trp179 in the WPD loop and BB allosteric pathway (*Wiesmann et al., 2004*) in the other direction (*Figure 5—figure supplement 4*). Thus, surface residues such as Tyr152 may be conformationally coupled to the buried underside of the WPD loop via a similar mechanism as BB binding – remotely modulating the Trp179 anchor via coordinated hydrophobic shifts – but from a different angle of attack, via Tyr176. Overall, such coordinated local shifts within the hydrophobic core likely 'lubricate' the transition between discrete global states of PTP1B.

## Assessing the ligandability of the surface of PTP1B using automated crystallography

Although the results described above establish a conformationally coupled network within the structure of PTP1B, allosteric inhibition also requires binding sites for small molecules that can conformationally bias this network to modulate function. To identify potential allosteric ligand-binding sites in PTP1B, we mapped the small-molecule binding potential or 'ligandability' of the entire protein surface. Specifically, we used small-molecule fragments, which by virtue of their small size provide a relatively large sampling of drug-like chemical space (*Murray and Blundell, 2010*). Astex Pharmaceuticals has previously explored fragment-based drug design for PTP1B (*Hartshorn et al., 2005*); however, that screen used molecules pre-selected to enrich for binders to phosphatase active sites, which contrasts with our goal of exploring the surface outside of the active site. To determine cocrystal structures of hundreds of fragments with PTP1B, we used the high-throughput fragment-soaking and crystallographic pipeline available at Diamond Light Source (*Collins et al., 2017*) to individually soak 1918 apo PTP1B crystals with small-molecule fragments in DMSO from several curated libraries, and another 48 with just DMSO. We then used robotic sample handling to automatically collect complete X-ray datasets at 100 K (*Figure 6—source data 1*). Of the 1966 total soaks, 1774 yielded diffraction data that could be successfully processed. The data were generally high-resolution: the average resolution was 2.1 Å, 65% of resolutions were better than 2.0 Å, and 87% were better than 2.5 Å (*Figure 6A*, *Figure 6—source data 1*). The large number of datasets enabled us to use the new Pan-Dataset Density Analysis (PanDDA) algorithm (*Pearce et al., 2017*) to reveal bound fragments. PanDDA performs weighted subtractions of the 'background' electron density (computed from apo and unbound datasets) from each electron density map (*Figure 6B–C*). The optimal subtraction, chosen by a heuristic, yields electron density corresponding to the ligand-bound fraction of unit cells in the crystal.

Our PanDDA analysis of 1774 datasets revealed 381 putative binding events. We manually inspected each putative binding event, and were able to confidently model the fragment in atomic detail for 110 hits (*Figure 6D*). Overall, 12 different sites in PTP1B were observed to bind fragments (*Figure 6E*). These sites are structurally distinct from one another – that is, they share no residues in common, and fragments bound within different sites do not overlap with each other. They are also widely distributed across the protein surface. Twenty-five fragments bind to multiple sites, but promiscuous binding is not unexpected from such small fragments, and still provides valuable information about favorable binding poses in each site.

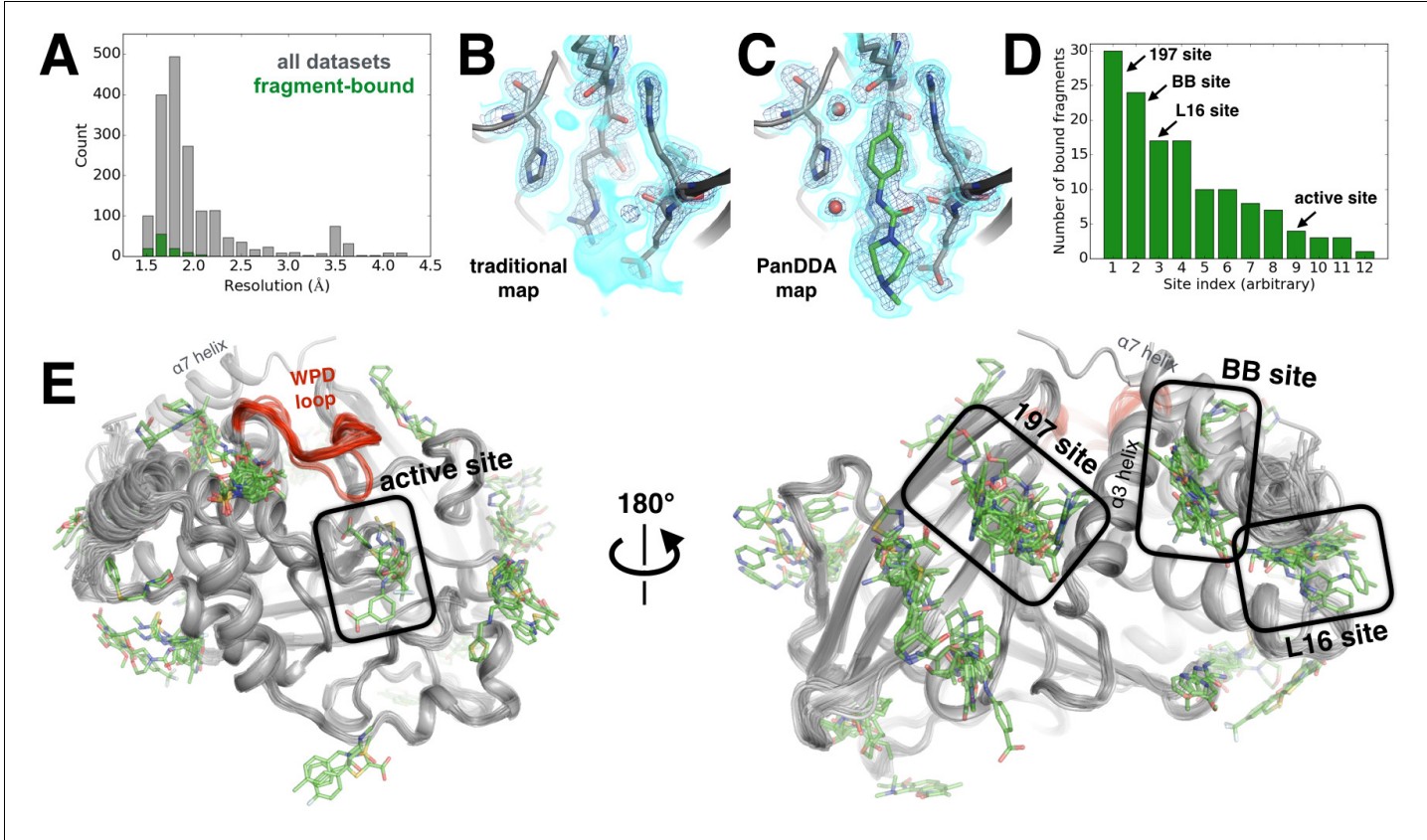

**Figure 6.** Electron-density background subtraction reveals small-molecule fragments at allosteric sites in PTP1B. (**A**) Histogram of X-ray resolution for 1774 structures of PTP1B soaked with small-molecule fragments (gray) vs. the 110 structures from that set with small-molecule fragments bound to PTP1B (green). (**B**) For one example fragment, a traditional 2Fo-Fc map contoured at 1.25 σ (cyan volume) and at 3.5 σ (blue mesh) provides no clear evidence for a bound fragment. (**C**) By contrast, a background-subtracted PanDDA event map (85% background subtraction in this case) contoured at the same levels clearly reveals the precise pose of the bound fragment, plus additional ordered water molecules that accompany it (red spheres). (**D**) PanDDA analysis and manual inspection reveal 110 fragment-bound structures of PTP1B, with bound fragments clustered into 12 non-overlapping binding sites. Some structures contain multiple bound copies of the same fragment. Several sites of interest are labeled. (**E**) Overview of bound fragments across the PTP1B surface. *Left*: front of protein, facing active site (WPD loop open and closed conformations in red). *Right*: back of protein, facing several fragment-binding hotspots: the 197 site, BB site, and L16 site. The viewing orientation in **E**) (*left*) is as in *Figure 1A* ('front side' of PTP1B). The viewing orientation in **E**) (*right*) is as in *Figure 1B* ('back side' of PTP1B).

DOI: https://doi.org/10.7554/eLife.36307.024

The following video and source data are available for figure 6:

**Source data 1.** Results of all 1966 fragment and DMSO soaks into PTP1B crystals.

DOI: https://doi.org/10.7554/eLife.36307.026

**Figure 6—video 1.** Movie version of *Figure 6E*.

DOI: https://doi.org/10.7554/eLife.36307.025

PanDDA initially identified >80 putative binding events in the active site. Many of these can be attributed to movements of the WPD loop (*Figure 2*), often induced by oxidation of the catalytic Cys215, which is a natural regulatory mechanism (*van Montfort et al., 2003*). Apart from these protein events and other false positives, we observe four fragments bound in the active site. This number is relatively low likely because our libraries were not customized to bind to the highly charged active site of PTP1B, as was the case in the Astex study (*Hartshorn et al., 2005*).

To identify allosteric binders, we examined sites outside of the active site. Strikingly, we observed 24 bound fragments in the BB allosteric site (*Figure 7A*). The poses of many of these fragments overlap portions of the BB scaffold (*Figure 7A*, *Figure 7—figure supplement 1*). However, many of them also contain chemical groups that suggestively protrude in new directions from the BB scaffold (*Figure 7—figure supplement 1*). This retrospective result validates the idea that fragment

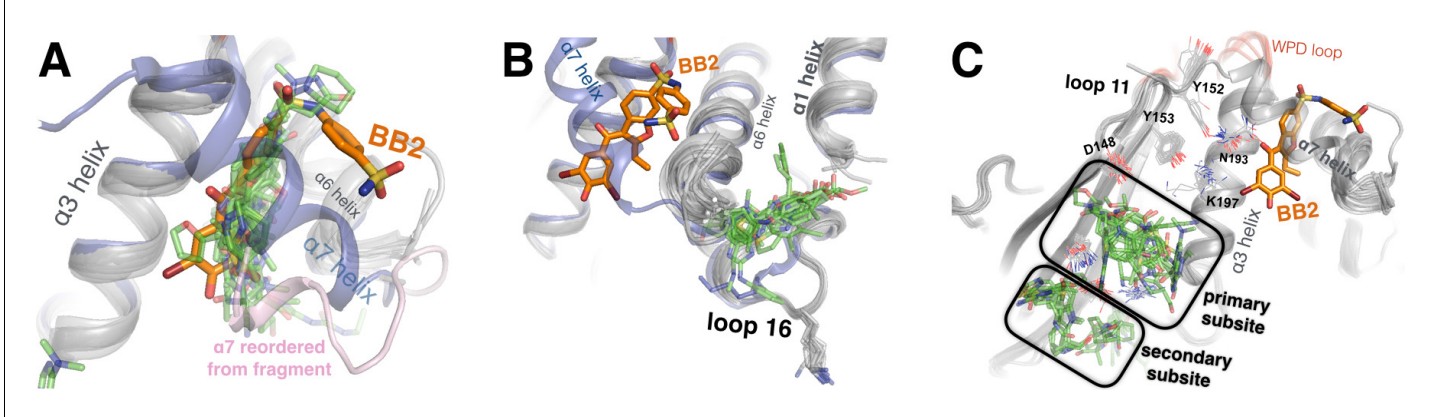

**Figure 7.** Fragments cluster at three binding hotspots distal from the active site. (**A**) Twenty-four fragments (green) bind to the same site and in similar poses as the BB2 inhibitor (orange, PDB ID 1t49), and similarly displace the α7 helix (foreground, transparent blue, PDB ID 1sug). BB2 is also shown in the following panels to emphasize that its binding site is distinct from the other fragment-binding hotspots. One structure with a fragment bound in this site features a reordered conformation of the α7 helix (pink). (**B**) Seventeen fragments bind to the L16 site, where they may modulate the conformations of loop 16, the α6 helix, and the protein's N-terminus on the α1 helix. (**C**) Thirty fragments bind to the 197 site in one primary subsite contacting K197, or a distinct secondary subsite nearby. The viewing orientation in (**A**) is as in *Figure 1B* ('back side' of PTP1B), except zoomed in on the BB site (labeled in *Figure 1B*). The viewing orientation in (**B**) is also as in *Figure 1B*, except looking left from the right of that image and zoomed in on the L16 and BB site site. The viewing orientation in A) is as also in *Figure 1B*, except zoomed in on the 197 site and BB site (labeled in *Figure 1B*). See also *Figure 6E* (right) for orientation.

DOI: https://doi.org/10.7554/eLife.36307.027

The following source data and figure supplements are available for figure 7:

**Source data 1.** Crystallographic statistics for fragment-bound structures.
DOI: https://doi.org/10.7554/eLife.36307.030

**Source data 2.** Small-molecule fragments tested in enzyme inhibition assays.
DOI: https://doi.org/10.7554/eLife.36307.031

**Figure supplement 1.** Fragments overlap with the BB allosteric inhibitor scaffold and suggest possible improvements.
DOI: https://doi.org/10.7554/eLife.36307.028

**Figure supplement 2.** Fragments in the 197 site overlay with glycerols from multitemperature structures.
DOI: https://doi.org/10.7554/eLife.36307.029

screening identifies binding sites, and specific fragment poses in those sites, that can be fruitfully exploited for allosteric inhibition. Interestingly, in one structure with a fragment bound in the BB site, the α7 helix adopts a reordered conformation that covers the binding site (*Figure 7A*), reminiscent of other examples in published structures and in our high-temperature datasets (*Figure 3—figure supplement 3*). These compounds could also inspire design of modified BB2 derivatives that may overcome the low affinity that limited the development of that series.

We also examined fragments bound to the L16 site and the 197 site, which were suggested to be allosteric sites by our multitemperature analysis of apo PTP1B. Excitingly, both sites are fragment-binding hotspots: 17 fragments bind to the L16 site (*Figure 7B*) and 30 fragments bind to the 197 site (*Figure 7C*). Thus, independent methods to assess allosteric coupling and ligandability converge on the same sites in PTP1B. Our results agree with previous studies, based on mutagenesis and NMR, which implicated several residues in the L16 site (*Cui et al., 2017*) and in the 197 site (*Choy et al., 2017*; *Cui et al., 2017*) as participating in an active-site-linked allosteric network. We also add value to those studies in another way: by reporting the binding poses of a few dozen small-molecule ligands that bind to these sites in atomic detail. Because these two sites are both conformationally coupled to the active site and capable of binding a variety of small molecules, they may be promising sites to explore for small-molecule allosteric inhibition of PTP1B activity.

The L16 site is between loop 16 (L16), the beginning of α1, and the end of α6. Most of the 17 fragments that bind here appear to 'pry apart' these elements (*Figure 7B*) to create a cryptic binding site (*Cimermancic et al., 2016a*). Because the end of α6 is coupled to the beginning of α7, which is perhaps the central allosteric hub of PTP1B (*Choy et al., 2017*), this site seems promising for

allosteric inhibition. The fragments that bind here are diverse but have some common features: aromatic moieties sandwich between Pro239 (of L16) and Met282 ($\alpha$6), and carboxyl groups hydrogen-bond to the backbone amide of Glu2 ($\alpha$6). These fragments do not spatially overlap with any fragments in the nearby BB site, confirming that the L16 site is genuinely distinct from the previously explored allosteric site.

The 197 site is on the opposite side of the BB site, near $\alpha$3 (including Lys197) and L11 (including Tyr152). Thirty fragments bind in the 197 site, with 14 in the primary subsite near Lys197, and 17 in a nearby but distinct subsite separated by a 'ridge' formed by the Gln157 and Glu170 sidechains (*Figure 7C*) (one fragment binds in both the primary subsite and the secondary subsite). These fragments are characterized by packing of aromatic moieties above Leu172, with additional aromatic or polar extensions in various directions. As with the L16 site, fragments in this site do not overlap with any fragments in the nearby BB site. However, several of the fragments in the 197 site do overlap with the positions of ordered glycerols from our multitemperature structures (which were absent from all fragment-soaked structures to avoid competition for binding) (*Figure 7—figure supplement 2*). Similarly, glycerol in PDB ID 3qkp and $\beta$-octylglucoside in PDB ID 2cmc (among other examples) bind to sites that are occupied by fragments in our structures. These findings emphasize that fortuitous binding of buffer components and other miscellaneous compounds can in some cases provide useful information about binding sites (*Mattos and Ringe, 1996*). It may be possible to link fragments in the primary subsite and secondary subsite to increase binding affinity. Although some fragments in the secondary subsite are largely stabilized by crystal-lattice contacts, they still enjoy favorable interactions with the protein that could potentially be useful for fragment extension. By contrast, the primary subsite is generally free from crystal-lattice contacts.

To assess the effect of the bound fragments on the structure of PTP1B more globally, for each dataset we built an ensemble structure consisting of both the ground state and the bound state. Each dataset was modeled with an innovative PDB format as a multiconformer structure that represents both a heterogeneous apo state and a heterogeneous holo state. Due to limitations in the PDB model format and in the ability of conventional refinement programs to interpret and create reasonable restraints for this model type, either one conformation or four alternative conformations were used to describe each residue, often when only two were necessary. Due to this forced degeneracy, refinement of coordinates, occupancy, and B-factors must be highly restrained. We interpret the resulting occupancies as a good approximation of the fraction of unit cells that have a ligand present. Refining these ensemble structures using restraints that avoid overfitting allowed for some structural differences between the two states to emerge. In principle, these structural differences could give some prediction of the functional effects one might expect upon developing a higher affinity version of the molecule. The refined ensemble structures were of high quality (*Figure 7—source data 1*). However, generally speaking, the structural differences were subtle: the global backbone RMSD (N, C$\alpha$, C atoms) between the ground state and bound state ranged from 0.7 to 1.7 Å. Cases with larger RMSD (>1.25 Å) generally involved either active-site fragments that directly shift the WPD loop, or fortuitous oxidation of the active-site Cys215 (*van Montfort et al., 2003*). Thus, fragment binding did not dramatically shift PTP1B from the open to the closed state in many of these structures. Many of these fragments are certainly benign binders that bind to non-allosteric sites. However, the strong preference for the open state even with fragments that bind to allosteric sites is likely due to the absence of glycerol, which is present in our multitemperature structures (see Materials and methods). It is likely that weak fragments do not overcome this energetic preference, and instead elicit conformational changes primarily in their immediate vicinity. Including glycerol to place the protein in a regime in which the open vs. closed states are more nearly isoenergetic during fragment soaks could potentially interfere with fragment binding to the 197 site, since ordered glycerols also fortuitously bind there (*Figure 7—figure supplement 2*).

## Validating a functional allosteric linkage with covalent tethering

The small-molecule fragments described above were identified by a naive screen and are not optimized for high-affinity binding to the 197 site or L16 site of PTP1B. Nevertheless, we selected 20 fragments that were deemed to bind in either site during early rounds of iterative PanDDA analysis (see Materials and methods) (*Figure 7—source data 2*) and tested whether they have allosteric effects using enzyme activity assays. Unsurprisingly, we did not observe inhibition of enzyme activity by the fragments up to the maximum concentrations we were able to assay due to solubility of the

fragments. It is important to note that this is not surprising due to the fragments' small sizes, relatively simple chemical structures, and low affinities (soaking with fragments at 30–150 mM concentrations resulted in observed occupancies of only 10–30% in the crystal structures). However, looking ahead, the dozens of cocrystal structures with small-molecule fragments bound at these promising allosteric sites (and at the previously explored BB2 site) that we have reported here offer a foothold for future medicinal chemistry efforts to design allosteric inhibitors for PTP1B.

Instead, here we focus on an alternative strategy to validate the concept of allosteric inhibition at the 197 site: covalent binding to enhance ligand occupancy. Specifically, we used 'Tethering' (*Erlanson et al., 2000*; *Erlanson et al., 2004*), in which a residue near the site of interest is mutated to cysteine, then the mutant is mixed with disulfide fragments under partially reducing conditions. Affinity of the fragments for the site of interest drives the formation of a disulfide bond between the fragment and the adjacent cysteine. The extent of cysteine labeling can be measured using whole-protein mass spectrometry, and serves as a metric to rank the affinity of fragments for a given site. One major advantage of Tethering over other fragment-based approaches is that it can leverage low-affinity binding events into quantitatively labeled protein species, whose enzymatic activity can then be assayed. Here, we used Tethering to successfully identify a covalent allosteric inhibitor at the 197 site of PTP1B (*Figure 8*).

For the allosteric 197 site in PTP1B, we chose to tether to a K197C mutant for several reasons. First, K197 is on the $\alpha$3 helix, which is a key allosteric element in PTP1B (*Choy et al., 2017*). We predicted that small-molecule tethering to our site could could perturb the helix via K197C, perhaps mimicking the effects of a free molecule binding to the WT protein and altering the K197 conformational distribution. Second, K197 and E200 are the two residues on $\alpha$3 whose C$\alpha$-C$\beta$ vectors point in roughly the correct direction toward the allosteric 197 site we describe; however, E200 engages in crystal-lattice contacts which would interfere with tethering in our P3$_1$21 space group, so we focused on K197C instead.

To efficiently explore the chemical space of covalent small molecules for the 197 site, we used a library of 1600 disulfide-capped fragments designed for covalent tethering experiments (*Kathman et al., 2014*; *Burlingame et al., 2011a*). From our initial screen, we identified 50 fragments that tethered to K197C > 3 standard deviations above the average percent tethering for all 1600 compounds (*Figure 8—figure supplement 1A*). We next measured the ability of these top fragments to modulate PTP1B's phosphatase activity (*Figure 8—figure supplement 1B*). Formation of the tethered complex followed by a pNPP assay identified only one fragment, **1** (*Figure 8—figure supplement 1C*), that appeared to inhibit PTP1B at a percentage comparable to the percentage of tethered complex (*Figure 8—figure supplement 1B*), suggesting a direct relationship between labeling and inhibition. While **1** thus showed the behavior we desired, the percent labeling and inhibition were relatively low. We hypothesized that altering the linker between the fragment core and the disulfide bond may lead to improved interactions between the protein and small molecule. For this reason, we designed and synthesized **2** (*Figure 8A*), which has the orientation of the amide bond reversed, allowing for one less carbon in the disulfide linker (*Figure 8—figure supplement 1C*).

When assayed, **2** showed improved tethering and inhibition of K197C relative to **1**. **2** exhibited dose-dependent tethering and partial noncompetitive allosteric inhibition of K197C with a tethering EC$_{50}$ of 7.8 ± 1.1 μM and a K$_i$ for pNPP activity of 7.1 ± 1.1 μM (maximum inhibition of ~60%) (*Figure 8B* and *Figure 8—figure supplement 2A*). Importantly, **2** appeared to show little to no tethering of WT* and minimal inhibition, supporting that **2**'s activity is specific to the 197 site and not due to tethering of the active-site cysteine found in both K197C and WT/WT* (*Figure 8C* and *Figure 8—figure supplement 2B*). In fact, the inhibition that is observed for WT* does not correlate with tethering, suggesting the inhibition may be from nonspecific factors, such as aggregation, at higher concentrations of **2** ($\geq$50 μM). Michaelis-Menten kinetic analysis of K197C in the presence of **2** (50 μM) showed a statistically significant ~50% reduction in V$_{max}$ relative to DMSO treatment, but no significant effect on K$_M$ for the pNPP substrate (*Figure 8—figure supplement 2C,E,F*). This supports a noncompetitive allosteric mechanism of inhibition. The effect on WT* kinetics was similar to the nonspecific inhibition observed in the dose titration experiment (*Figure 8—figure supplement 2D*), once again supporting that the activity of **2** is specific for the K197 site on PTP1B. To further profile the inhibitory effect of tethering of **2** on K197C, we assayed the ability of the tethered complex to dephosphorylate the alternative substrate DiFMUP (*Welte et al., 2005*). As with pNPP, the

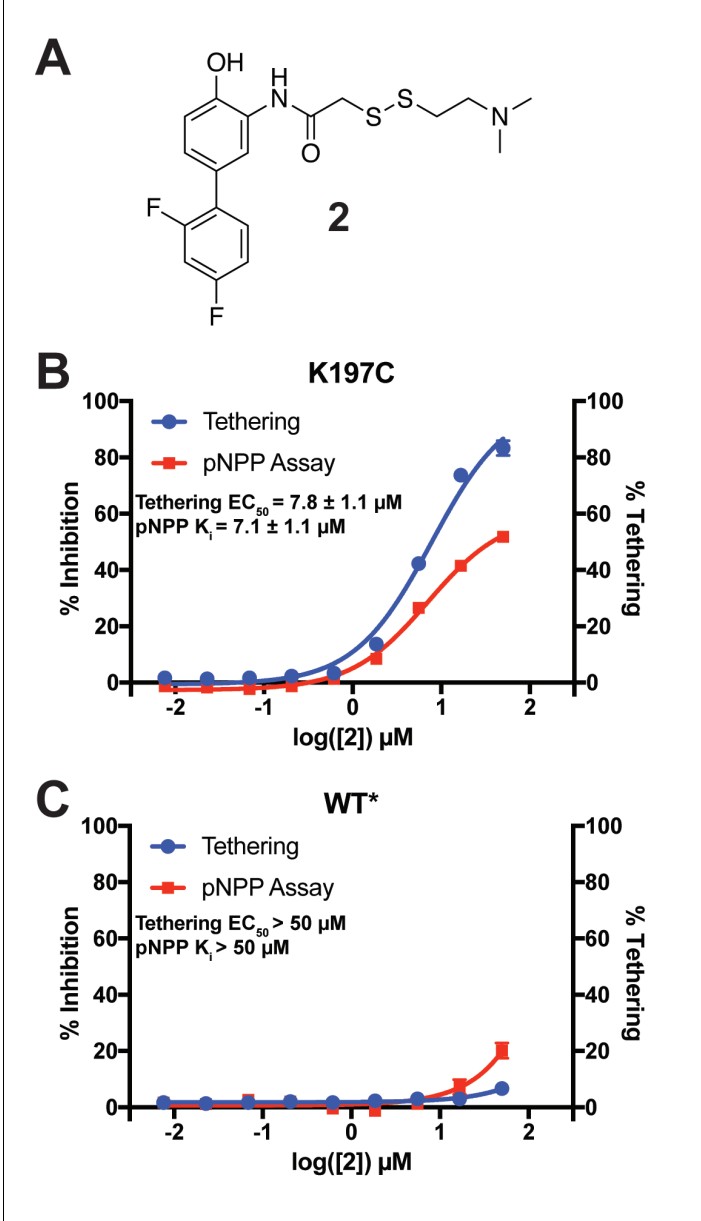

**Figure 8.** Characterization of a functional covalent allosteric inhibitor. (**A**) The chemical structure of our covalent disulfide fragment **2**. (**B**) Tethering and Inhibition of K197C at varying concentrations of **2**. The tethering $EC_{50}$ observed was $7.8 \pm 1.1$ μM and the $K_i$ for pNPP activity was $7.1 \pm 1.1$ μM with a maximum inhibition of ~60%. Tethering data represents all tethering events combined. (**C**) Tethering and Inhibition of WT* at varying concentrations of **2**. The tethering $EC_{50}$ and the $K_i$ for pNPP activity were both >50 μM. Tethering data represents all tethering events combined. Data represent the mean of three independent assays ± the standard error of the mean. All assays were performed in the presence of 100 μM of β-mercaptoethanol.

DOI: https://doi.org/10.7554/eLife.36307.032

The following figure supplements are available for figure 8:

**Figure supplement 1.** Identification of a functional covalent allosteric inhibitor.

DOI: https://doi.org/10.7554/eLife.36307.033

**Figure supplement 2.** **2** tethers to only a single cysteine in K197C and inhibits through a reduction in $V_{max}$.

DOI: https://doi.org/10.7554/eLife.36307.034

**Figure supplement 3.** **2** inhibits K197C catalysis of DiFMUP through a reduction in $V_{max}$.

DOI: https://doi.org/10.7554/eLife.36307.035

tethered complex was inhibited, with kinetic analysis showing a dramatic reduction in $V_{max}$, but no significant effect on $K_M$ (*Figure 8—figure supplement 3*). These results once again support a partial noncompetitive allosteric mechanism of inhibition.

To futher validate that **2** acts specifically through the K197 site and to explore the mechanism of inhibition by **2**, we solved a high-resolution (1.95 Å, *Table 1*) crystal structure of K197C tethered with **2**. The structure confirms that **2** tethers to K197C rather than to the active-site catalytic Cys215, and also that tethered **2** resides in the 197 site (*Figure 9A*) rather than in the relatively nearby BB site, which is also theoretically within reach of the tethering linker on the other side of the α3 helix. We modeled **2** as partially populated and, indeed, the 83% refined occupancy in the crystal structure was very similar to the ~85% conjugation measured after tethering in solution prior to crystallization. **2** adopts a conformation in which the two rings are nearly coplanar. This interpretation is further validated by a 'polder map', in which both the ligand and bulk solvent are omitted (*Liebschner et al., 2017*) (*Figure 9—figure supplement 1*). While coplanar biphenyl rings are typically believed to be disfavored due to steric clashes, it is possible that hydrogen bonding of D148 with the phenol combined with the electronegativity of the para-fluoro leads to delocalization of the rings' electrons and promotes a coplanar conformation. Additionally, **2** packs against the hydrophobic floor, centered on Leu172, of the relatively shallow binding pocket in the 197 site. Trapping of coplanar biphenyl rings covalently attached to a protein has previously been reported (*Pearson et al., 2015*).

To elucidate conformational changes induced by **2**, we also solved a high-resolution (1.95 Å, *Table 1*) crystal structure of apo K197C in the same crystal form for comparison. As mentioned previously, PTP1B remains in the open state without glycerol; glycerol was absent from the tethered K197C structure (to avoid competition for binding in the 197 site) and from the K197C apo structure (for consistency), so we are unable to see any dramatic shifts in the global open-closed equilibrium that **2** may induce. Tyr153 shifts its position slightly and Tyr152 responds by shifting fully to its up rotamer, but this is likely due to the loss of interactions with the WT K197 upon mutation. Beyond these mutation-induced effects, we see some conformational changes associated with tethering of **2**. The key residue Asn193 (*Choy et al., 2017*) changes rotamers, the sidechain of Phe196 on α3 'slides' to change its aromatic stacking arrangement with Phe280 on α6 (*Figure 9C*), and Glu276 – which contacts the wedge residue Leu192 (*Choy et al., 2017*) – rotates sidechain dihedral angles. These sidechain movements appear to couple to subtle, more distributed backbone shifts (*Deis et al., 2014*) of the α3 helix, several residues of which move up toward the WPD loop by ~0.5 Å (*Figure 9C*). Interestingly, these sidechain and particularly backbone movements are somewhat similar to those between the two macrostates of apo PTP1B at high temperature (*Figure 9D*). Thus, although the mechanistic details remain unclear, allosteric inhibition by **2** may involve conformational changes, especially of α3, that are similar to those that occur during the global transition from the open to the closed state (*Choy et al., 2017*). This interpretation is consistent with a recent report that mutations (Y153A, M282A) in what we here recognize as the 197 site and L16 site alter the equilibrium between the WPD loop's open and closed states (*Cui et al., 2017*). We note that the noncompetitive allosteric mechanism observed suggests that tethering **2** to K197C may shift the protein's energy landscape in such a way as to alter the kinetics of WPD loop motions. Future work to explore this issue would nicely complement the crystallographic and functional analysis we provide here.

Interestingly, several of the other noncovalent fragments bound to WT* overlay well with the aromatic rings of **2** tethered to K197C (*Figure 9B*). This structural convergence suggests promising new avenues for future medicinal chemistry efforts. First, more conservatively, portions of specific fragments could be added to **2** to yield improved covalent allosteric inhibitors for K197C PTP1B. Second, perhaps more promisingly, portions of **2** could be combined with (portions of) specific fragments to create potent new non-covalent allosteric inhibitors for WT PTP1B.

## Discussion

Our analysis of PTP1B paints a portrait of an inherently allosteric system. Allostery is fundamentally tied to protein functions such as catalysis via the theme of conformational motions (*Goodey and Benkovic, 2008*). Here, we have harnessed new approaches in X-ray crystallography to map coordinated conformational redistributions that underlie allostery in the dynamic enzyme PTP1B.

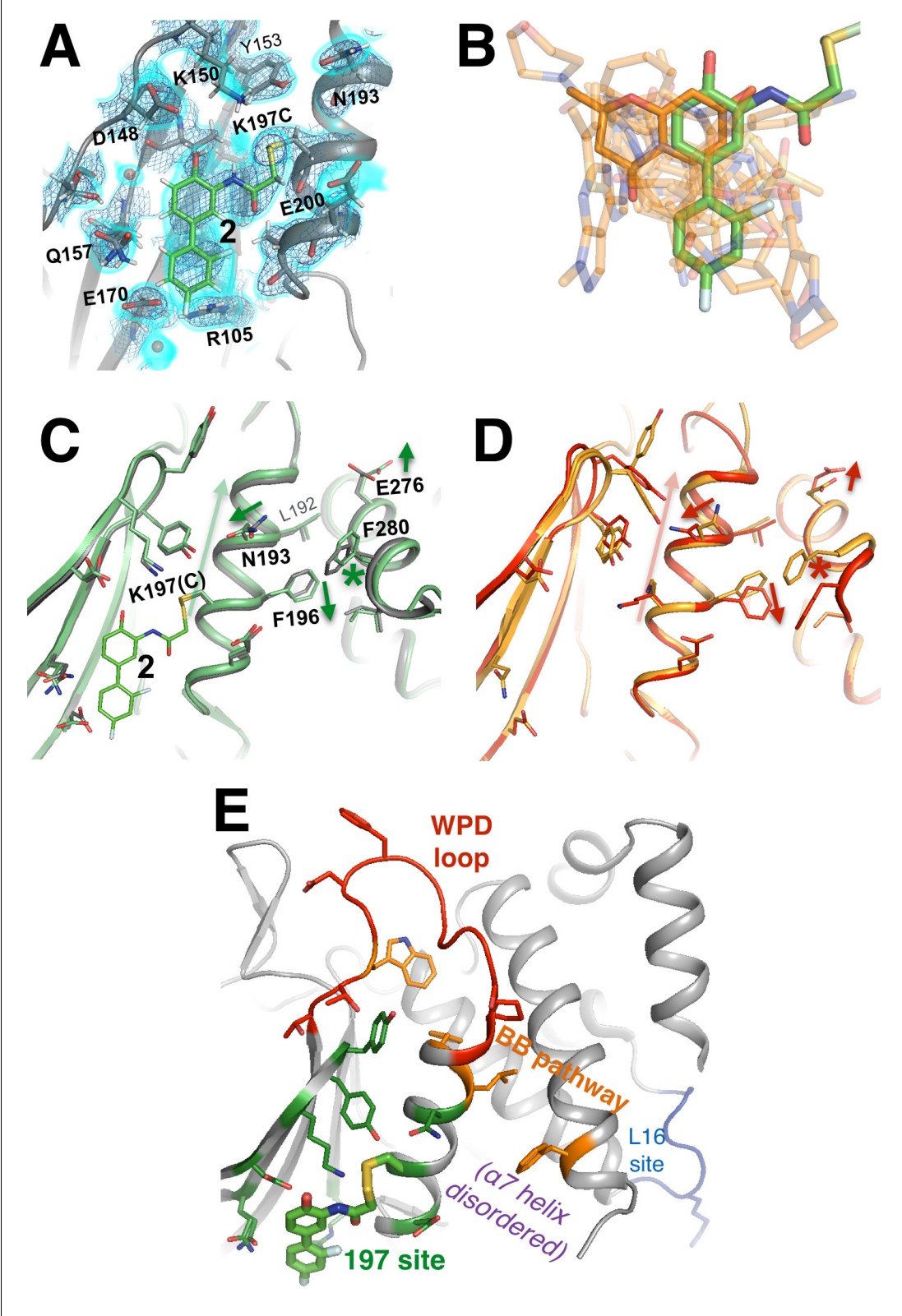

**Figure 9.** A functional small-molecule inhibitor tethered to the allosteric 197 site. (**A**) The tethered inhibitor **2** is highly ordered (~85% occupancy) in the 197 site, as seen by 2Fo-Fc electron density for our 1.90 Å structure contoured at 0.75 σ (cyan volume) and at 1.5 σ (blue mesh) that is continuous to the K197C sidechain. A few waters (transparent red spheres) which appear to be mutually exclusive with the molecule are likely displaced by binding. (**B**) Many fragments from WT* cocrystal structures (transparent orange) overlay well with **2** in the K197C cocrystal structure (green). One fragment in

*Figure 9 continued on next page*

*Figure 9 continued*

particular (solid orange) has a ring substructure that overlays very closely with a ring substructure of **2**. (**C**) The K197C **2**-tethered structure (green) is similar to the K197C apo structure (gray), but upon tethering there are several conformational changes (arrows and asterisk) in the α3 helix: the whole backbone shifts up in this view slightly leading back into the WPD loop (top), N193 switches rotamers, and the sidechains of F196 and E200 move within rotameric wells. The end of the α6 helix, including E276 and F280, appears to respond in concert. (**D**) Several of these changes mirror changes from open-to-closed apo PTP1B (arrows and asterisk) as seen in the two conformations of our 278 K model (red/orange). (**E**) Overview as in *Figure 2E* for context. The viewing orientation in (**A**) is as in *Figure 1B* ('back side' of PTP1B), except zoomed in on the 197 site (labeled in *Figure 1B*). The viewing orientation in (**C–E**) is as in *Figure 1B*.

DOI: https://doi.org/10.7554/eLife.36307.036

The following video and figure supplement are available for figure 9:

**Figure supplement 1.** Polder omit map of K197C tethered structure.

DOI: https://doi.org/10.7554/eLife.36307.037

**Figure 9—video 1.** Movie version of *Figure 9A*.

DOI: https://doi.org/10.7554/eLife.36307.038

Metaphorically, we were able to use this map of PTP1B's 'intramolecular nervous system' to reveal allosteric 'pressure points' that enable long-range modulation of its function.

Proteins sample many conformations from a complex energy landscape (*Frauenfelder et al., 1991*), many of which are accessible and represented among the millions to trillions of molecules in a protein crystal. However, an X-ray crystallographic dataset provides only ensemble-averaged information – so it is difficult to decipher individual minor conformations from a single dataset. A key to our work was harnessing the power of *en masse* structural analysis, which let us reveal minor conformations and the shifts between them that allow a dynamic protein to function. We exploited families of structures in two different ways. First, we contrasted structures at several different temperatures (*Keedy et al., 2015b*) for PTP1B to track coordinated conformational shifts which underlie allosteric communication. Second, we used hundreds of structures of PTP1B with different small-molecule fragments to calculate a statistical 'background' electron density map representing the unbound state, which we could subtract to reveal fragment-bound conformations (*Pearce et al., 2017*) for many allosteric sites. This requires using the PDB format of alternative locations to encode both compositional and conformational heterogeneity within a single model. Our multi-structure equilibrium X-ray approaches complement other methods for breaking the degeneracy of ensemble-averaged data to resolve multiple conformations of macromolecules. For example, 3D classification algorithms in cryo-electron microscopy enable in silico purification of different compositional and conformational states (*Scheres, 2016*). Time-resolved X-ray experiments, for example with free-electron lasers, offer great promise for mapping conformational changes with both spatial and temporal resolution, although general experimental strategies are still forthcoming for the vast majority of proteins that are non-photoactivatable (*Hekstra et al., 2016*). More generally, integrative modeling algorithms can synthesize data from disparate sources at different resolutions, including solution NMR or small-angle X-ray scattering, to build ensembles of structures that are consistent with all the experimental data (*van den Bedem and Fraser, 2015*; *Russel et al., 2012*).

By exploiting a new multitemperature multiconformer X-ray approach, we have identified a collective allosteric network that is contiguous on the 'back side' of the protein, centered around the quasi-ordered α7 helix (*Figure 2E*, *Figure 9E*). This network includes the BB site, which was previously targeted with a small-molecule allosteric inhibitor (*Wiesmann et al., 2004*). It also includes adjacent sites (the 197 site and the L16 site) in either direction from the BB site, which have not been targeted previously with small-molecule inhibitors. Several residues in these additional sites were implicated as being part of putative allosteric sites by recent work using mutagenesis and NMR chemical shift and dynamics information (*Choy et al., 2017*; *Cui et al., 2017*). Our work agrees with those studies in identifying the 197 site and L16 site as potentially important players in PTP1B's collective allosteric network. We additionally complement them by revealing, in atomic detail, alternative conformations that these sites natively populate. Our work suggests that allosteric perturbations do not necessarily induce conformational changes in PTP1B – instead, the alternative conformations are already latently sampled by the apo protein and are simply stabilized by the allosteric perturbations. Portions of the allosteric network we observe here in PTP1B – in particular the series of aromatic and hydrophobic residues linking the allosteric BB site to the active site (*Figure 3*)

and the cluster of aromatic residues behind and beneath the WPD loop (*Figure 5—figure supplement 4*) – are analogous to the dynamic hydrophobic spines that play central roles in allosteric activation of protein kinases (*Taylor and Kornev, 2011*; *Kim et al., 2017*). Our analysis suggests that allostery in PTP1B is characterized by interdependent conformational changes spanning length scales: helical order-disorder transitions, hydrophobic shifts, local sidechain rotamer changes, subtle helical twists, and large discrete active-site loop motions.

Our work reveals new opportunities for long-range control of PTP1B function by impinging upon tendrils of this expanded allosteric network with small molecules. Specifically, we have used two different small-molecule techniques with complementary strengths to reveal new footholds for developing allosteric inhibitors for PTP1B in the future. First, we used a new high-throughput method with small-molecule fragments to map the ligandability of the entire surface of PTP1B with high structural resolution. Although individually these fragments have low affinity, collectively the 110 protein:fragment structures we report reveal overlapping poses with a multitude of precise binding interactions at specific sites in PTP1B (*Figure 7A,B,C*). Many of the 11 binding sites outside the active site are likely to be benign. Importantly, our multitemperature X-ray analysis of the apo protein provides a way to predict which binding sites are instead likely to be allosterically coupled to function. Based on the result that the same small number of sites are both (a) implicated as allosteric by multitemperature X-ray analysis of conformational changes in apo PTP1B and (b) the most ligandable sites from the fragment screen across the entire surface, we conclude that conformational changes may be important for allosteric ligand binding in this protein. Second, we used covalent tethering to probe the functional effects of a high-occupancy ligand at one promising allosteric site. Our work takes advantage of a synergy between fragments and tethering. Fragments bind weakly, but allow for the visualization of hundreds of chemical entities bound to distinct sites in a protein. By contrast, tethering does not provide high-throughput structural information, but allows for targeted perturbations to functionally probe a specific site, and additionally can provide lower-throughput structural information about chemical matter that allosterically inhibits a protein variant. Importantly, the structural and functional information from these techniques can be combined to open doors for future structure-based drug design efforts (*Figure 9B*). Together, our results illuminate a promising new region of chemical space that may be fruitfully explored by future medicinal chemistry efforts to develop potent non-covalent allosteric inhibitors for WT PTP1B.

Although our work helps set the stage for such efforts, generating potent and selective ligands based on linking of fragment-screening hits is a difficult and time-consuming process that is beyond the scope of this current study. While none of the fragments identified to bind at the 197 site in our work have the potency or selectivity for proper assaying of their effect on enzyme activity, let alone for use as a cell-active ligand, their identification supports the ligandability of the 197 site, and motivates future efforts to identify potent ligands at this site. Our use of covalent tethering at the 197 site allowed us to drive occupancy of fragment **2** at this site and functionally validates the allosteric effect of ligand binding at this site on PTP1B phosphatase activity. The percent tethering observed for **2** is correlated with the percent inhibition observed (*Figure 8B*), indicating that the tethered compound drives inhibition. However, the maximum inhibition is only ~60% (*Figure 8B* and *Figure 8—figure supplement 3A*), which is consistent with a partial noncompetitive allosteric mechanism of inhibition. This mechanism is supported by our kinetic analysis of **2**-bound K197C (*Figure 8—figure supplement 2C* and *Figure 8—figure supplement 3B*) where the reduction in enzyme rate is driven by a reduction in $V_{max}$, not a change in $K_M$ (*Figure 8—figure supplement 2E,F* and *Figure 8—figure supplement 3C,D*). Partial inhibition is a common paradigm in noncompetitive allosteric inhibition, where ligand occupancy does not perfectly correlate with inhibition (*Ramsay and Tipton, 2017*; *Whiteley, 2000*).

The 'druggability' of the 197 site in WT PTP1B is further supported by recent work using structure-based simulations and virtual screening, which found that the 197 site bound several diverse molecules in silico, and was the most responsive to allosteric perturbations among several potential allosteric pockets identified in PTP1B (*Kumar et al., 2018*). Together with our findings, these data imply that while the development of potent non-covalent allosteric inhibitors targeting the 197 site may be arduous, a concerted effort toward this goal may ultimately be fruitful.

Future efforts to develop potent non-covalent allosteric inhibitors for PTP1B can explore an additional feature of the allosteric 197 site that may enhance its druggability. As stated above, the quasi-disordered α7 helix is capable of reordering into different conformations, some of which cover the

α3-α6 region including the 197 site. We observe several reordered α7 conformations under different conditions: with BB3 at 273 K, with BB1 at 100 K, in the S295F (α7) mutant, in the L192A (α3) mutant with an active-site inhibitor (*Figure 3—figure supplement 3C*), and with a fragment in the BB site (*Figure 7A*). Similarly, the disordered C-terminus, including α7, accommodates a recently reported allosteric inhibitor (*Krishnan et al., 2014*). For the BB site, the malleability of α7 is likely a double-edged sword. On one hand, reordered conformations of α7 may contribute binding energy to allosteric inhibitors by forming a 'lid' over what is partially a 'cryptic site' (*Cimermancic et al., 2016*). On the other hand, they may also accommodate different small-molecule variants equally well, such that it is difficult to improve upon inhibition – that is, the structure-activity relationship (SAR) is flat. However, the 197 site is structurally distinct from the BB site, and is accessible to different, more C-terminal portions of α7. Future work will test the hypothesis that the 197 site can yield improved allosteric inhibitors that take advantage of these unique structural features. Notably, it is possible that the flat SAR observed at the BB site is due to an inhibition mechanism that is unrelated to the particular binding site, which is not uncommon in PTP1B drug discovery efforts – if this is the case, targeting the 197 site may face similar hurdles.

Our work motivates specific future efforts to allosterically inhibit PTP1B activity for therapeutic purposes. However, the allosteric network illuminated here by probing PTP1B with non-biological perturbations (temperature and small molecules) in vitro may also be relevant to how the enzyme is regulated in vivo, in at least three ways. First, in addition to directing subcellular localization (*Frangioni et al., 1992*), the quasi-disordered C-terminal region of PTP1B reorders to interact with the 'back side' of the catalytic domain in different ways, is phosphorylated at disordered serine residues in vivo to regulate function (*Brautigan and Pinault, 1993*), and mediates allosteric inhibition by natural product molecules (*Krishnan et al., 2014*). Moreover, removing α7 reduces activity in PTP1B (*Choy et al., 2017*), and removing the disordered C-terminus in the related phosphatase TCPTP reduces activity even more dramatically (*Hao et al., 1997*). Second, Tyr152 (*Figure 5*) is phosphorylated in vivo, which contributes to binding to the insulin receptor kinase (IRK) (*Bandyopadhyay et al., 1997*; *Rhee et al., 2001*) and is required for binding to N-cadherin (*Bandyopadhyay et al., 1997*; *Rhee et al., 2001*). Tyr152 phosphorylation is sterically compatible only with the 'up' rotamer, which is correlated with the global open state of PTP1B; it is therefore possible that Tyr152 phosphorylation and/or subsequent IRK or cadherin binding events directly affect PTP1B activity via the allosteric network we report here. Third, in several other protein tyrosine phosphatases (PTPs), protein-protein interactions occur on the 'back side' of the catalytic domain – coinciding with where we observe three major allosteric sites in PTP1B. For example, in RPTPγ and RPTPε, the non-catalytic D2 domain binds to the catalytic D1 domain at an interface coinciding with the allosteric network we report on the back of PTP1B (*Barr et al., 2009*). Additionally, the N-terminus of PTPL1 wraps around the area coinciding with the L16 site in PTP1B, and docks in the area coinciding with the α7 helix in PTP1B (*Villa et al., 2005*). Together, these observations suggest that the allosteric network we establish here within the catalytic domain of PTP1B may function as a 'receiver' for allosteric inputs from the C-terminus in cells. If so, this strategy operates in parallel with other mechanisms such as active-site oxidation (*van Montfort et al., 2003*) and phosphorylation of other sites in the catalytic domain (*Ravichandran et al., 2001*) as part of a complex, multifaceted regulatory scheme.

Finally, it is interesting to note that different protein tyrosine phosphatases (PTPs) share a structurally conserved catalytic domain with PTP1B – but have different variants of the α7 helix or even entirely different N- or C-terminal domains (*Alonso et al., 2004*) that can be trapped in inhibitory conformations for allosteric inhibition, as recently realized for SHP2 (*Chen et al., 2016*). Similarly, regulatory domains or subdomains were recently targeted for allosteric inhibition of the serine/threonine phosphatases Wip1 (*Gilmartin et al., 2014*) and PP1 (*Carrara et al., 2017*). It will be exciting to dissect the mechanisms by which different PTPs are allosterically controlled by their specific regulatory domains – both to unravel these proteins' unique cellular roles, and to reveal new opportunities to correct their dysregulation in disease.

## Materials and methods

### Cloning, expression, and purification

For all 'wild-type' PTP1B experiments here, we used what we refer to as the WT* construct: residues 1–321, with the C32S/C92V double mutation (*Erlanson et al., 2003*) to prevent off-target tethering reactions, in a pET24b vector with a kanamycin resistance gene. K197C, K197A, Y152G, and Y153A were created using site-directed mutagenesis from the WT* construct.

Protein was expressed and purified as previously reported (*Pedersen et al., 2004*), with some minor variations. For expression, we transformed BL21 *E. coli* cells with plasmid, grew cells on LB + kanamycin plates overnight at 37°C, inoculated 5 mL starter cultures of LB + kanamycin with individual colonies, grew shaking overnight at 37°C, inoculated larger 1 L cultures of LB + kanamycin, grew shaking at 37°C until optical density at 600 nm was approximately 0.6–0.8, induced with 100 mM IPTG, and grew shaking either for 4 hr at 37°C or overnight at 18°C. Cell pellets ('cellets') were harvested by centrifugation and stored at −80°C in 50 mL conical tubes.

For purification, we first performed cation exchange with an SP FF 16/10 cation exchange column (GE Healthcare Life Sciences) in lysis buffer (100 mM MES pH 6.5, 1 mM EDTA, 1 mM DTT) with a multi-stage 0–1 M NaCl gradient (shallow at first for elution of PTP1B, then steeper); PTP1B eluted around 200 mM NaCl. We then performed size exclusion with a Superdex 75 size exclusion column (GE Healthcare Life Sciences) in size exclusion buffer (100 mM MES pH 6.5, 1 mM EDTA, 1 mM DTT, 200 mM NaCl). PTP1B appeared highly pure in SDS-PAGE gels.

### Crystallization

WT* PTP1B was dialyzed into crystallization buffer (10 mM Tris pH 7.5, 0.2 mM EDTA, 25 mM NaCl, 3 mM DTT) with at least a 200x volume ratio overnight at 4°C. We were unable to grow apo WT* PTP1B crystals initially, so we synthesized the active-site inhibitors OBA and OTP as in (*Andersen et al., 2002*) (OBA = compound **3a**, OTP = compound **12h**). We were unable to solubilize OTP as used in (*Pedersen et al., 2004*). Instead, we co-crystallized PTP1B with OBA (*Andersen et al., 2000*). We first solubilized OBA to 250 mM in DMSO, then created a 10:1 molar ratio of PTP1B:OBA. Crystallization drops were set in 96-well sitting- or hanging-drop format at 4°C with 10–15 mg/mL protein with 1 μL of protein solution + 1 μL of well solution (0.2–0.4 M magnesium acetate, 0.1 M HEPES pH 7.3–7.6, 12–17% PEG 8000), then trays were incubated at 4°C. Crystals several hundred μm long grew within a few days, and often continued to grow bigger for several more days. We created seed stocks from these crystals by pipetting the entire drop into 50 μL of well solution, iterating between vortexing for 30 s and sitting on ice for 30 s several times, and performing serial 10-fold dilutions in well solution. Apo crystals were grown by introducing seed stock (0x, 10x, or 100x diluted) into freshly set drops, either by streaking with a cat whisker or pipetting a small amount (e.g. 0.1 μL into a 2 μL drop). Serial seeding using new apo crystals successively improved crystal quality. We also added ethanol to the well solution based on an additive screen (Hampton Research), and added glycerol to mimic the previously published apo structure protocol (*Pedersen et al., 2004*), resulting in the following final WT* PTP1B crystallization well solution: 0.3 M magnesium acetate, 0.1 M HEPES pH 7.5, 0.1% β-mercaptoethanol, 16% PEG 8000, 2% ethanol, 10% glycerol. The resulting crystals were used for our WT* PTP1B multitemperature analysis.

We also crystallized WT* PTP1B in MRC SwissCi 3-well sitting-drop trays. Protein was at 30–50 mg/mL protein in the same crystallization buffer. The well solution was very similar except for having a slightly lower precipitant concentration (13–14% PEG 8000) and no glycerol. Drops were set at room temperature with 135 nL protein solution + 135 nL well solution + 30 nL seed stock, then trays were incubated at 4°C. Crystals appeared within a few days. The best seed stocks had been diluted 10-100x. These crystals were used for the BB3-soaking and fragment-soaking experiments.

We crystallized apo K197C in the microbatch format with Al's oil covering all wells. Protein was at 5–30 mg/mL in the same crystallization buffer as WT* but without DTT. The well solution was 0.3 M magnesium acetate, 0.1 M HEPES pH 7.5, 0.1% β-mercaptoethanol, 10–26% PEG 8000, 2% ethanol. Drops were set on ice with 1 μL protein solution + 1 μL well solution, then trays were incubated at 4°C. Crystals appeared within a few days. We also grew apo K197C crystals in a few other similar conditions.

We crystallized K197C tethered to **2** in the 96-well hanging-drop format. Protein was at 15 mg/mL in the same crystallization buffer as WT* but without DTT. The well solution was 0.2 M magnesium acetate tetrahydrate, 20% PEG 3350. Drops were set at room temperature with 100 nL protein solution + 100 nL well solution, then trays were incubated at 4°C. Crystals appeared within a few days.

## X-ray data collection

We used PDB ID 1sug for the apo WT 100 K dataset. The apo WT* 278 K dataset was collected at Stanford Synchrotron Radiation Lightsource (SSRL) beamline 12–2. All fragment-soaked datasets were collected at Diamond Light Source beamline I04-1. All other datasets were collected at Advanced Light Source (ALS) beamline 8.3.1.

For apo WT* 180, 240, and 278 K, crystals had been grown in 2 µL drops with 10% glycerol in the mother liquor, then 1.5 µL of 50% glycerol was added several hours before data collection, resulting in a final concentration of ~27% glycerol. Some crystals were also dabbed into more 50% glycerol just before mounting.

For BB3-complexed WT*, crystals were soaked with 125 nL of 10 mM BB3 (in DMSO). No glycerol was present in these crystals.

For apo WT* and BB3-complexed WT*, crystals were looped and placed in a plastic capillary with ~70% mother liquor,~30% water to prevent dehydration during data collection, regardless of temperature; datasets were obtained at different temperatures simply by adjusting the cryojet temperature before placing the crystal on the goniometer. Helical data collection (translation along the crystal coupled to goniometer rotations) was used to expose fresh regions with each shot, to minimize radiation damage.

For apo and **2**-tethered K197C, crystals were simply looped and directly mounted on the goniometer in front of the cryojet. For apo K197C, a small amount of ice was likely present on the crystal.

For fragment-soaked PTP1B, WT* PTP1B crystals in MRC SwissCi 3-well sitting-drop trays were soaked with small-molecule fragments using acoustic droplet ejection technology and a database mapping individual fragments to individual crystals as described (*Collins et al., 2017*). PTP1B crystals were quite tolerant to DMSO, so we were able to achieve high fragment concentrations and long incubation times: we soaked overnight for >8 hr at final concentrations of 30% DMSO and 30–150 mM fragment (depending on the concentration of the fragments in the source library). Additionally we collected X-ray data for 48 'apo' datasets (soaked with DMSO), 42 of which gave high-resolution datasets, to better establish the unbound background electron density for PanDDA analysis. Despite the high DMSO concentrations, we did not observe difference electron density consistent with any ordered DMSO molecules bound to PTP1B. Some fragments were soaked into additional crystals if good datasets were not obtained from the initial soak; however, only 2 of the 110 fragment-bound datasets contain the same fragment. We also collected a relatively small number of trial datasets (28) near room temperature instead of cryogenic temperature, but they were generally low-resolution, and none revealed bound fragments.

Most crystals stuck to the bottoms of wells regardless of construct and tray format, but it was often possible to gently dislodge them, or to physically break them off then expose the unperturbed portion of the crystal to the X-ray beam. Each dataset in this study was collected from a single crystal.

## X-ray data processing

To process the multitemperature and tethered datasets, we used XDS (*Kabsch, 2010*). In each case we chose a resolution cutoff for which CC1/2 (*Karplus and Diederichs, 2012*) was statistically significant at the 0.1% level (above 0.4). We created a new set of $R_{free}$ flags for the 278 K WT* apo dataset, then transferred them to the MTZ file of every other dataset with the reflection file editor in PHENIX (*Adams et al., 2010*) (for PDB ID 1sug, we first deleted the existing $R_{free}$ flags). We solved each structure by molecular replacement with Phaser (*McCoy et al., 2007*). One solution was obtained for each dataset. For WT*, we used PDB ID 1c83 with all waters and the WPD loop removed. For K197C, we used a refined WT* PTP1B model for molecular replacement.

For fragment-soaked datasets, we used XDS and a custom script [80; copy archived at https://github.com/elifesciences-publications/xds_iter] to automatically determine resolution cutoffs for all

datasets. The resolution cutoff was initialized at 1.4 Å and incremented until the following criteria were met for the highest-resolution bin: at least 1.0 I/σ(I), at least 50% CC1/2 (*Karplus and Diederichs, 2012*), and at least 90% completeness. $R_{free}$ flags were created for the highest-resolution dataset by transferring and extending the flags from the 278 K WT* apo dataset using the PHENIX reflection file editor. These $R_{free}$ flags were then transferred from that highest-resolution dataset to every other dataset in the fragment-soaking experiment. For PanDDA to accept the MTZ files as inputs, it was necessary to modify each file so that all columns (H, K, L, structure factors, map coefficients, and R-free flags) had the same number of indices; no observations were omitted in this step. We then phased each dataset with Phaser using a reference model that was created by interpreting a high-resolution DMSO-soaked apo dataset. Next, we refined each initial model from Phaser using phenix.refine with the following flags to prevent excessive coordinate drift: 'reference_coordinate_restraints.enabled=True 'reference_coordinate_restraints.sigma=0.1'. Structure factors were then dropped from MTZ files, leaving map coefficients as inputs to PanDDA. Filled map coefficients (from PHENIX) were used to avoid Fourier series truncation effects in PanDDA maps. The resulting models were used as input to PanDDA (see below).

## Structure modeling

For *Figure 2—figure supplement 1*, we re-refined the following 36 structures of PTP1B from the PDB either as-is, or with a dual-conformation WPD loop: 1bzj 1kak 1oem 1oeo 1sug 1t48 1t49 1t4j 2azr 2b07 2bgd 2f6f 2f6t 2f6v 2f6w 2f6y 2f6z 2f70 2f71 2h4k 2hb1 2qbp 2qbq 2qbr 2qbs 2zmm 2zn7 3cwe 3d9c 3eax 3eb1 3eu0 3i7z 3i80 3sme 4i8n. For the dual-conformation refinements, we constrained occupancies of the open + closed conformations of the WPD loop to 1.

For multitemperature WT/WT*, we refined the initial model using phenix.refine for 10 macrocycles with automated water picking turned off. Next, we inserted preliminary open and closed alternative conformations for the WPD loop, and refined for another 10 macrocycles with automated water picking turned on. Finally, we performed several rounds of manual rebuilding, including manual addition and deletion of protein, solvent, and glycerol conformations and refinement with automated water picking turned off. Anisotropic B-factors were not used in refinement. The α7 helix was modeled as alternative conformation A only, with the unmodeled B conformation presumed to correspond to the disordered state; this allowed the occupancy of the ordered state to be refined. It was necessary to provide explicit occupancy parameter files to phenix.refine in some cases. For many residues, conformations obtained from PDB ID 1sug or 1t49 or another of our datasets (usually higher temperatures) were useful for 'filling in' missing density. Often the missing conformations would not have been obvious based on the map alone, but once inserted and refined they seemed to fit well. This cross-dataset conformational-sampling approach also had the effect of emphasizing differences between models from different temperatures while minimizing differences due to chance or arbitrary modeling choices. Nevertheless, we encourage future users of these datasets to compare across different temperatures based at least in part on the electron density, and not just our models. The building process was guided by all-atom structure validation with MolProbity (*Chen et al., 2010*). The 100 K WT model (1sug) is truly WT, whereas our new WT* datasets are all C32S/C92V, as noted above; however, both cysteine-scrubbing mutations are structurally conservative, distal to the active site, and apparently uncoupled from the WPD loop and all allosteric regions.

Glycerol (~20% final) was present in WT* crystals during each multitemperature data collection to maintain consistency with the 100 K WT structure (PDB ID 1sug), in which glycerol was used as a cryoprotectant. Several ordered glycerol molecules, including those in contact with the closed WPD loop and at the allosteric 197 site, were evident from electron density at all temperatures. However, in some cases, it was difficult to differentiate between ordered waters, glycerols, or simply noise in the map. For example, the electron density was uncertain at some of the elevated temperatures for some glycerols originally modeled in PDB ID 1sug. When glycerol was omitted from crystals, the WPD loop was entirely in the open conformation regardless of temperature, from cryogenic temperature to near room temperature (data not shown). Our interpretation is that ordered glycerols in the active site, which are evident from the electron density at all temperatures, make weak contacts with the WPD loop's closed conformation, and thus shift the protein's energy landscape to a regime in which the open vs. closed conformations are near enough to isoenergetic that temperature can modulate their populations. This interpretation is strengthened by the fact that these glycerol

molecules align well with a bound mimic of the pTyr substrate (PDB ID 1pty), which causes the loop to close during the catalytic cycle.

For K197C, we used a similar refinement procedure, including many manual tweaks of alternative conformations for protein and water atoms. For the **2**-tethered K197C structure, we omitted **2** for many rounds of refinement, allowing the electron density for the missing molecule to become extremely convincing before we finally added it to the model. The distance between the sulfur atoms in K197C and the ligand was restrained to 2.15 Å with a σ of 0.1 Å for refinement.

For fragment-soaked datasets, we used the PanDDA approach (*Pearce et al., 2017*) in a few stages. First, using PanDDA version 0.1, we ran pandda.analyse, and interpreted and modeled events. Next, using the new PanDDA version 0.2, we ran pandda.analyse again. During this second run, datasets which were events in the first run were excluded from background density calculation, and datasets that had substantial map artifacts or very noisy/low-quality maps in the first run were excluded entirely.

Some events for which we modeled bound fragments in the earlier PanDDA version 0.1 runs were not detected as events in the final PanDDA version 0.2 run. In these cases, we manually created event maps based on the 1-BDC background subtraction values from the earlier PanDDA runs. In most cases, visual inspection confirmed that these early events likely correspond to bound fragments.

Many PanDDA 'events' in the active site corresponded not to ligand binding, but rather to protein conformational changes of the WPD loop, P-loop, and Y46 loop that are caused by oxidation of the catalytic Cys215, a natural PTP1B regulatory mechanism (*van Montfort et al., 2003*). Some other active-site events were difficult to interpret, perhaps due to active-site dynamics or differences in the appropriate background model for the open vs. closed state of the protein; future methodological improvements may clarify modeling in such cases.

We built a generic unbound-state model by interpreting both an average map for the highest-resolution bin and one of the best individual apo datasets. The WPD loop was modeled as open, Tyr152 was modeled with two alternative sidechain conformations on the loop 11 backbone that is compatible with the open WPD loop, and the N-terminus (start of α1) and C-terminus (end of α6, since α7 was disordered) were fit as well as possible. Ordered waters were also manually positioned. This generic unbound-state model was superposed onto each PanDDA input model in the correct reference frame, then refined, to create an unbound-state model for each dataset.

For each fragment-bound state, we inspected the fragment binding site, plus the several interesting regions of the protein mentioned above, in detail interactively. Waters were copied over from the unbound-state model, then moved or deleted where they conflicted with the bound fragment and/or the PanDDA event map. For a small number of planar fragments, several copies of the fragment bind in a parallel stack bridging the 197 site and a symmetry-related copy of the WPD-loop residue Phe182 via a crystal-lattice contact. Some areas such as Tyr152 were modeled with alternative conformations in the bound state only when they were well justified in the event map; otherwise we generally adhered to the unbound-state model. The correct modeling choice for the termini was uncertain in some cases.

To refine structures for the 110 datasets with one or more modeled fragments, first we created restraints files for the ligands with eLBOW (*Moriarty et al., 2009*). For a small number of ligands, we additionally used AceDRG (*Long et al., 2017*) and found that AceDRG generated more realistic restraints. Next, the pandda.export method in PanDDA version 0.2 was used to create an 'ensemble structure' containing both the unbound state (including alternative conformations) and the bound state (including alternative conformations) in one multiconformer model. In pandda.export, the parameter 'options.prune_duplicates_rms = 0.2' was used to merge alternative conformations that were highly similar, and the parameter 'duplicates.rmsd_cutoff = 0.4' was used to restrain the coordinates of somewhat similar alternative conformations to be identical. These parameter values were chosen to effectively merge residues with very similar coordinates, while still allowing residues we evaluated as having genuine alternative conformations to remain separate and unrestrained. The resulting geometry restraint files from pandda.export are necessary to minimize overfitting or coordinate drift during refinement of this model type.

For refinement of the ensembles representing multiconformer models of the apo and bound states, we first refined each ensemble structure with phenix.refine to obtain water positions. The first stage of PHENIX refinement was 10 macrocycles with no removal or addition of waters

('ordered_solvent = False') to let the existing waters relax into local minima. The second stage of PHENIX refinement was another 10 macrocycles with automated removal and addition of waters ('ordered_solvent = True') to remove waters that were unable to reach local minima and add waters that were clearly missing. Adding and removing waters, when compared to only removing them, generally had negligible effect on MolProbity scores, but improved $R_{work}$ and $R_{free}$. During this first PHENIX refinement stage to obtain water positions, occupancies were fixed to the original PanDDA BDC value for the ground state and 1-BDC for the bound state; occupancy was distributed evenly between substates when the ground state or the bound state had alternative conformations for some residues. We observed coordinate drift and unstable B-factors for the protein with PHENIX refinement. Therefore, we copied the water positions obtained from PHENIX into the initial ensemble models, and refined with Refmac (*Murshudov et al., 2011*). To do so, we first set all B-factors to 40 Å$^2$, set bound-state occupancies to 2*(1-BDC) and unbound-state occupancies to 1–2*(1-BDC) (with occupancy evenly distributed across alternative conformations within each state), and generated new restraints files that included the water molecules by running the PanDDA utility giant.make_restraints with the same RMSD parameter as with giant.merge_conformations: 'duplicates.rmsd_cutoff = 0.4'. Then, each ensemble model was refined with Refmac using giant.quick_refine with the ligand CIF and custom giant.make_restraints restraint parameter files using the protocol herein. First, each model was refined for the default 10 cycles, with the extra arguments to Refmac 'MAKE HOUT Yes', to preserve hydrogens, and 'HOLD 0.001 100 100' to restrain XYZ coordinates but still allow for some geometry regularization and encourage B-factor and occupancy convergence. Next, the output from that refinement was fed into a loop of Refmac refinement with the default 10 cycles per run, and the 'HOLD 0.0001 100 100' argument, essentially fixing the XYZ coordinates, while letting occupancies and B-factors refine. Our output was the result of the 4th round (1 round + 3 rounds) of refinement in Refmac. However, the occupancies refined with these refinements did not converge to the correct occupancy (as seen by huge difference peaks describing the ligand). We then refined these structures with PHENIX, fixing the XYZ coordinates and manually scanning across possible occupancies while refining B-factors with the following settings: refinement.refine.strategy = individual_adp, hydrogens.optimize_scattering_contribution = False, main.number_of_macro_cycles = 10, optimize_mask = True, optimize_adp_weight = True. While in principle one could interpret the difference density to pick an optimal refined occupancy, no other statistics calculated provided a clear choice of occupancy. We ultimately chose to deposit occupancies of the bound state at 2.2 times the event map occupancy (1-BDC). This occupancy choice was motivated by the trend previously found by *Pearce et al. (2017)*. In cases where the total bound occupancy was 50% or higher, the models were manually inspected, and a few dropped to low occupancies that minimized difference features of the ligand. The resulting final ensemble structures of the unbound state plus the fragment-bound state were converted from PDB to mmCIF format and deposited in the PDB using the new multimodel submission procedure.

## Visualization

Coot (*Emsley et al., 2010*) was instrumental to visualizing and interactively modeling all structures. PyMOL (*Schrödinger, 2016*) was used for all molecular graphics after initial modeling. We frequently used the volume rendering feature for low-contour electron density alongside the traditional mesh for higher-contour electron density.

## Synthesis of tethered compounds

### Synthesis of 3-amino-2',4'-difluoro-[1,1'-biphenyl]—4-ol

A solution of 2,4-difluorophenyl boronic acid (188 mg, 1 mmol, 1.0 equiv), 2-amino-4-bromophenol (7.2 mg, 0.62 mmol, 1.2 equiv), Pd(PPh$_3$)$_4$ (57.8 mg, 0.05 mmol, 0.05 equiv), and Na$_2$CO$_3$ (318 mg, 3 mmol, three equiv) in THF (8 mL) was stirred at 90 °C overnight. The reaction was allowed to cool, taken up in water, and then extracted 3x with EtoAC. Organic layers were combined, washed with brine, dried over Na$_2$SO$_4$, concentrated *in vacuo* and purified using flash chromatography (MeOH/DCM) to obtain 22.1 mg of product (10% yield). Calcd for C$_{12}$H$_{10}$F$_2$NO (M+H$^+$): 222.07; Found 222.8.

## Synthesis of 2

To a mixture of 3-amino-2',4'-difluoro-[1,1'-biphenyl]–4-ol (22.1 mg, 0.1 mmol, one equiv), dithiodi-glycolic acid (9.1 mg, 0.05 mmol, 0.5 equiv), HOBt·$H_2O$ (19.9 mg, 0.13 mmol, 1.3 equiv), and DIPEA (0.226 µL, 1.3 mmol, 1.3 equiv) in THF (300 µL), EDCI·HCl (25 mg, 0.13 mmol, 1.3 equiv) was added. The reaction was stirred at room temperature overnight. To this was added a solution of bis[2-(N,N-dimethylamino)ethyl]disulfide dihydrochloride (70.3 mg, 0.25 mmol, 2.5 equiv) and TCEP (2.5 mg, 0.01 mmol, 0.01 equiv) in THF/$H_2O$ (8:3, 275 µL). The reaction was stirred at room temperature over-night. The reaction was taken up in EtoAC/$H_2O$, and extracted three times with EtOAc. Organic layers were combined, washed with brine, dried over $Na_2SO_4$, concentrated in vacuo and purified using flash chromatography (DCM/MeOH) followed by preparative HPLC ($C_{18}$ column (50 × 19 mm), Methanol/Water–0.05% formic acid gradient: 10:90 to 100:0 over 12 min; 20 mL/min; 254 nm detec-tion for 18 min.) to obtain 9.1 mg of **2** (23% yield). $^1$H NMR (400 MHz, Acetone-$d_6$) δ 9.55 (s, 1H), 8.17 (s, 1H), 8.06–8.04 (m, 1H), 7.54–7.45 (m, 1H), 7.22–7.19 (m, 1H), 7.12–7.00 (m, 3H), 3.94–3.78 (m, 2H), 3.15–2.99 (m, 4H), 2.54–2.52 (m, 6H). Calcd for $C_{18}H_{21}F_2N_2O_2S_2$ (M+H$^+$): 399.1; Found 398.93.

## Tethering

We screened K197C against a previously synthesized library of 1600 disulfide fragments made avail-able by the UCSF Small Molecule Discovery Center (SMDC) (*Kathman et al., 2014*; *Burlingame et al., 2011b*).

For the screen, tethering reactions were performed using the following conditions: 1x tethering buffer (25 mM Tris pH 7.5, 100 mM NaCl), with 500 nM of K197C, 1 mM β-mercaptoethanol, and 100 µM of fragment (0.2% DMSO), 1 hr at rt. Unless otherwise noted, tethering reactions for follow-up experiments and activity assays were performed using the following conditions: 1x tethering buffer, 1 µM of K197C, 0.1 mM β-mercaptoethanol, and 50 µM of fragment (2% DMSO), 1 hr at rt. For DiFMUP assays 100 µM of fragment (0.2% DMSO) was used during tethering. For crystallogra-phy, tethering reactions were performed using the following conditions: 1x tethering buffer, 0.76 mg/mL of K197C, 0.1 mM β-mercaptoethanol, 500 µM of TCS401, and 250 µM of fragment (2% DMSO), 2 hr at rt. A total reaction size of 3.5 mL was used for preparation of crystallography sam-ples. Following labeling, the reaction was dialyzed into crystallization buffer overnight to remove TCS401 and unbound fragment. In all cases, the percent of tethering was measured using a Waters Xevo G2-XS Mass Spectrometer, and calculated by comparing the relative peak heights of the unmodified and modified protein. Tethering $EC_{50}$ values were calculated using nonlinear fitting in Prism 7 (Graphpad), n = 3.

## Activity assays

For activity assays of WT* PTP1B vs. allosteric mutants (*Figure 5—figure supplement 3*), protein was diluted to 269 nM (WT*) or 200 nM (mutants) in a variant of pNPP activity assay buffer (50 mM HEPES pH 7, 100 mM NaCl, 1 mM EDTA, and 1 mM DTT). WT* assays were performed at 269 nM protein and mutant assays were performed at 200 nM, so WT* data is normalized to 200 nM in both panels in *Figure 5—figure supplement 3*. Enzyme activity assays were performed across 10 p-nitro-phenyl phosphate (pNPP) concentrations obtained by serial two-fold dilutions starting from 20 mM. A no-enzyme well was also assayed. Absorbance at 405 nm for each reaction was monitored every 30 s for 5 min using a Tecan Infinite M200 Pro. The rate (mAU/min) of each reaction was calculated over the 5 min. Michaelis-Menten parameters were then calculated using Prism 7 (Graphpad). $k_{cat}$ values were calculated using an pNPP extinction coefficient of 18,000 M$^{-1}$ cm$^{-1}$ and a path length of 0.29 cm. These parameters for WT* PTP1B were similar to those reported previously for WT (*Choy et al., 2017*); small discrepancies may be due in part to differences in the length of the pro-tein construct being used.

For activity inhibition assays of WT* PTP1B with small-molecule fragments, 20 fragments were chosen early in the iterative PanDDA analysis process (see 'Structure modeling'). Protein was diluted to 200 nM in a variant of pNPP activity assay buffer (50 mM HEPES pH 7, 100 mM NaCl, 1 mM EDTA, 0.05% Tween-20, and 100 mM β-mercaptoethanol). Enzyme activity assays were performed with 0.15 or 1 mM fragment in 2% DMSO (final) or with 2% DMSO without fragment as a control, with 5 mM pNPP. A no-enzyme well was also assayed. Absorbance at 405 nm for each reaction was

monitored every 30 s for 5 min. The rate (mAU/s) of each reaction was calculated over the 5 min. These rates were compared with fragment vs. with DMSO.

For single-point assays of tethered K197C, completed tethering reactions (post 1 hr incubation) were diluted to a final concentration of 200 nM K197C with a variant of pNPP activity assay buffer (50 mM HEPES pH 7, 100 mM NaCl, 1 mM EDTA, 0.05% Tween-20, and 100 mM β-mercaptoethanol) and pNPP (5 mM final). A no-enzyme well and a DMSO-only well were also assayed. Absorbance at 405 for each reaction was monitored every 30 s for 5 min using a Tecan Infinite M200 Pro. Percent inhibition was calculated using the following equation: $100(1- ((Rate_{Fragment}-Rate_{No\ Enzyme})/(Rate_{DMSO}-Rate_{No\ Enzyme})))$.

For **2** titration assays of tethered K197C and WT*, tethering reactions were performed at nine different concentrations of **2** obtained by serial three-fold dilutions starting at 50 μM. After 1 hr incubation, the reactions were diluted to a final concentration of 100 nM K197C and WT* with a variant of pNPP activity assay buffer (50 mM HEPES pH 7, 100 mM NaCl, 1 mM EDTA, 0.05% Tween-20, and 100 mM β-mercaptoethanol), the same concentration of **2** as used during tethering, and pNPP (5 mM final). A no-enzyme well and a DMSO-only well were also assayed. Absorbance at 405 nm for each reaction was monitored every 30 s for 5 min using a Tecan Infinite M200 Pro. The rate (mAU/s) of each reaction was calculated over the 5 min. Percent inhibition was calculated using the following equation: $100(1- ((Rate_{Fragment}-Rate_{No\ Enzyme})/(Rate_{DMSO}-Rate_{No\ Enzyme})))$. $K_i$ values were calculated using nonlinear fitting in Prism 7 (Graphpad), n = 3.

For pNPP kinetics experiments with tethered complexes, completed K197C and WT* tethering reactions (post 1 hr incubation) were diluted to a final concentration of 200 nM K197C and WT* with a variant of pNPP activity assay buffer (50 mM HEPES pH 7, 100 mM NaCl, 1 mM EDTA, 0.05% Tween-20, and 100 mM β-mercaptoethanol), **2** (50 μM), and 12 pNPP concentrations obtained by serial two-fold dilutions starting from 20 mM. Absorbance at 405 nm for each reaction was monitored every 30 s for 5 min using a Tecan Infinite M200 Pro. The rate (mAU/min) of each reaction was calculated over the 5 min. Data was plotted and fit using Prism 7 (Graphpad), n = 3. $V_{max}$ and $K_M$ values were calculated using the Michaelis-Menten nonlinear fit in Prism 7 (Graphpad). All p-values and significances were calculated using a FDR-Approach Multiple t-test in Prism 7 (Graphpad).

For DiFMUP inhibition experiments with tethered complexes, completed K197C and WT* tethering reactions (post 1 hr incubation) were diluted to a final concentration of 2 nM K197C and WT* with a variant of DiFMUP activity assay buffer (50 mM HEPES pH 7, 100 mM NaCl, 1 mM EDTA, 0.05% Tween-20), and 9.4 μM of DiFMUP. Fluorescence at 450 nm with 358 nm excitation for each reaction was monitored every 30 s for 3 min using a Tecan Infinite M200 Pro. The rate (ΔF/min) of each reaction was calculated over the 3 min. Percent inhibition was calculated using the following equation: $100(1- ((Rate_{Fragment}-Rate_{No\ Enzyme})/(Rate_{DMSO}-Rate_{No\ Enzyme})))$, n = 3.

For DiFMUP kinetics experiments with tethered complexes, completed K197C tethering reactions (post 1 hr incubation) were diluted to a final concentration of 2 nM K197C with a variant of DiFMUP activity assay buffer (50 mM HEPES pH 7, 100 mM NaCl, 1 mM EDTA, 0.05% Tween-20), and 11 DiFMUP concentrations obtained by serial two-fold dilutions starting from 300 μM. Fluorescence at 450 nm with 358 nm excitation for each reaction was monitored every 30 s for 3 min using a Tecan Infinite M200 Pro. The rate (ΔF/min) of each reaction was calculated over the 3 min. Data was plotted and fit using Prism 7 (Graphpad), n = 3. $V_{max}$ and $K_M$ values were calculated using the Michaelis-Menten nonlinear fit in Prism 7 (Graphpad). All p-values and significances were calculated using a FDR-Approach Multiple t-test in Prism 7 (Graphpad).

## Data availability

Multiconformer models and structure factors for the multitemperature WT and WT* (6B90, 6B8E, 6B8T, 6B8X), BB3-bound (6B8Z), K197C apo (6BAI) and tethered (6B95) datasets have been deposited in the Protein Data Bank (*Berman et al., 2000*).

We have made publicly available several files that document our PanDDA analysis of all WT* fragment-soaked datasets. For each dataset, we provide a model of the unbound state, structure factors, an average map for the corresponding resolution bin, a PanDDA Z-map, and one or more PanDDA event map(s) as applicable. For fragment-bound datasets, we also provide the refined ground state model and the bound state model (before they were merged into an ensemble and refined) as separate PDB files, along with PHENIX, Refmac, and ligand restraint files used in the ensemble refinement. Additionally, we provide an overall PanDDA log file. These files are hosted at

Zenodo at the following DOI: 10.5281/zenodo.1044103. Finally, using a new deposition procedure, refined ensemble structures for the 110 WT* fragment-bound datasets have been deposited to the PDB (5QDE, 5QDF, 5QDG, 5QDH, 5QDI, 5QDJ, 5QDK, 5QDL, 5QDM, 5QDN, 5QDO, 5QDP, 5QDQ, 5QDR, 5QDS, 5QDT, 5QDU, 5QDV, 5QDW, 5QDX, 5QDY, 5QDZ, 5QE0, 5QE1, 5QE2, 5QE3, 5QE4, 5QE5, 5QE6, 5QE7, 5QE8, 5QE9, 5QEA, 5QEB, 5QEC, 5QED, 5QEE, 5QEF, 5QEG, 5QEH, 5QEI, 5QEJ, 5QEK, 5QEL, 5QEM, 5QEN, 5QEO, 5QEP, 5QEQ, 5QER, 5QES, 5QET, 5QEU, 5QEV, 5QEW, 5QEX, 5QEY, 5QEZ, 5QF0, 5QF1, 5QF2, 5QF3, 5QF4, 5QF5, 5QF6, 5QF7, 5QF8, 5QF9, 5QFA, 5QFB, 5QFC, 5QFD, 5QFE, 5QFF, 5QFG, 5QFH, 5QFI, 5QFJ, 5QFK, 5QFL, 5QFM, 5QFN, 5QFO, 5QFP, 5QFQ, 5QFR, 5QFS, 5QFT, 5QFU, 5QFV, 5QFW, 5QFX, 5QFY, 5QFZ, 5QG0, 5QG1, 5QG2, 5QG3, 5QG4, 5QG5, 5QG6, 5QG7, 5QG8, 5QG9, 5QGA, 5QGB, 5QGC, 5QGD, 5QGE, 5QGF).

## Acknowledgements

We thank Michelle Arkin for helpful suggestions; the Small Molecule Discovery Center at UCSF for use of the disulfide-fragment library; Chris Wilson, Ken Hallenbeck, and Gregory Lee for assistance while performing the Tethering screen; Nicholas Rettko for help generating mutant PTP1B plasmids; Nigel Moriarty and Brandi Hudson for help with covalent ligand restraints; Patrick Collins, Alice Douangamath, and Tobias Krojer for help operating the XChem fragment-screening pipeline; and Anil Verma for operation of the Research Center at Harwell crystallization facility. We also 'thank' LK for 'help' with data collection.

## Additional information

### Funding

| Funder | Grant reference number | Author |
| --- | --- | --- |
| A.P. Giannini Foundation | Postdoctoral Fellowship | Daniel A Keedy |
| Helen Hay Whitney Foundation | | Zachary B Hill |
| National Cancer Institute | K99CA203002 | Zachary B Hill |
| National Cancer Institute | F31 CA180378 | T Justin Rettenmaier |
| Wellcome Trust | | Frank von Delft |
| National Cancer Institute | CA191018 | James A Wells |
| Kinship Foundation | | James S Fraser |
| Pew Charitable Trusts | | James S Fraser |
| David and Lucile Packard Foundation | | James S Fraser |
| National Institute of General Medical Sciences | GM110580 | James S Fraser |
| National Science Foundation | STC-1231306 | James S Fraser |
| University of California | LFR-17-476732 | James S Fraser |
| National Institute of General Medical Sciences | GM123159 | James S Fraser |
| National Institute of General Medical Sciences | GM124169 | James S Fraser |
| National Institute of General Medical Sciences | GM124149 | James S Fraser |

The funders had no role in study design, data collection and interpretation, or the decision to submit the work for publication.

## Author contributions

Daniel A Keedy, Conceptualization, Data curation, Software, Formal analysis, Funding acquisition, Validation, Investigation, Visualization, Methodology, Writing—original draft, Writing—review and editing; Zachary B Hill, Conceptualization, Data curation, Formal analysis, Funding acquisition, Validation, Investigation, Visualization, Methodology, Writing—original draft, Writing—review and editing; Justin T Biel, Data curation, Software, Formal analysis, Validation, Investigation, Visualization, Methodology, Writing—original draft, Writing—review and editing; Emily Kang, Investigation, Methodology, Writing—review and editing; T Justin Rettenmaier, Conceptualization, Investigation, Methodology; José Brandão-Neto, Resources, Methodology, Writing—review and editing; Nicholas M Pearce, Conceptualization, Data curation, Software, Formal analysis, Validation, Methodology, Writing—review and editing; Frank von Delft, Resources, Supervision, Funding acquisition, Project administration, Writing—review and editing; James A Wells, Conceptualization, Resources, Supervision, Funding acquisition, Project administration, Writing—review and editing; James S Fraser, Conceptualization, Resources, Formal analysis, Supervision, Funding acquisition, Validation, Investigation, Visualization, Methodology, Writing—original draft, Writing—review and editing

## Author ORCIDs

Daniel A Keedy ⓘ http://orcid.org/0000-0002-9184-7586
Justin T Biel ⓘ http://orcid.org/0000-0002-0935-8362
José Brandão-Neto ⓘ https://orcid.org/0000-0001-6015-320X
Nicholas M Pearce ⓘ http://orcid.org/0000-0002-6693-8603
Frank von Delft ⓘ http://orcid.org/0000-0003-0378-0017
James A Wells ⓘ http://orcid.org/0000-0001-8267-5519
James S Fraser ⓘ http://orcid.org/0000-0002-5080-2859

## Decision letter and Author response

Decision letter https://doi.org/10.7554/eLife.36307.044
Author response https://doi.org/10.7554/eLife.36307.045

---

# Additional files

## Supplementary files

• Supplementary file 1. PDB deposition details for all structures reported.
DOI: https://doi.org/10.7554/eLife.36307.039

• Transparent reporting form
DOI: https://doi.org/10.7554/eLife.36307.040

## Data availability

Data have been deposited in PDB under the accession codes: 6B90, 6B8E, 6B8T, 6B8X, 6B8Z, 6BAI, 6B95, 5QDE, 5QDF, 5QDG, 5QDH, 5QDI, 5QDJ, 5QDK, 5QDL, 5QDM, 5QDN, 5QDO, 5QDP, 5QDQ, 5QDR, 5QDS, 5QDT, 5QDU, 5QDV, 5QDW, 5QDX, 5QDY, 5QDZ, 5QE0, 5QE1, 5QE2, 5QE3, 5QE4, 5QE5, 5QE6, 5QE7, 5QE8, 5QE9, 5QEA, 5QEB, 5QEC, 5QED, 5QEE, 5QEF, 5QEG, 5QEH, 5QEI, 5QEJ, 5QEK, 5QEL, 5QEM, 5QEN, 5QEO, 5QEP, 5QEQ, 5QER, 5QES, 5QET, 5QEU, 5QEV, 5QEW, 5QEX, 5QEY, 5QEZ, 5QF0, 5QF1, 5QF2, 5QF3, 5QF4, 5QF5, 5QF6, 5QF7, 5QF8, 5QF9, 5QFA, 5QFB, 5QFC, 5QFD, 5QFE, 5QFF, 5QFG, 5QFH, 5QFI, 5QFJ, 5QFK, 5QFL, 5QFM, 5QFN, 5QFO, 5QFP, 5QFQ, 5QFR, 5QFS, 5QFT, 5QFU, 5QFV, 5QFW, 5QFX, 5QFY, 5QFZ, 5QG0, 5QG1, 5QG2, 5QG3, 5QG4, 5QG5, 5QG6, 5QG7, 5QG8, 5QG9, 5QGA, 5QGB, 5QGC, 5QGD, 5QGE, 5QGF and further data available at https://zenodo.org/record/1044103

The following dataset was generated:

| Author(s) | Year | Dataset title | Dataset URL | Database, license, and accessibility information |
|---|---|---|---|---|
| Keedy, Biel, Fraser | 2017 | PanDDA analysis of PTP1B screened against fragment libraries | https://doi.org/10.5281/zenodo.1044103 | Publicly available at Zenodo (https://zenodo.org) |

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
