## [Decision Letter]

[Editors’ note: a previous version of this study was rejected after peer review, but the authors submitted for reconsideration. The first decision letter after peer review is shown below.]

Thank you for submitting your work entitled "New routes for PTP1B allosteric inhibition by multitemperature crystallography, fragment screening and covalent tethering" for consideration by *eLife*. Your article has been reviewed by three peer reviewers, one of whom is a member of our Board of Reviewing Editors, and the evaluation has been overseen by a Senior Editor. The reviewers have opted to remain anonymous.

Our decision has been reached after consultation between the reviewers. Based on these discussions and the individual reviews below, we regret to inform you that your work will not be considered further for publication in *eLife*.

This manuscript investigates PTP1B through a combination of multiple-temperature X-ray crystallography and structure determination of hundreds of individual small-molecule fragments bound to PTP1B. The amount of work described is impressive and the experiments appear to be well executed. The authors noted that there are a number of allosteric inhibitors already described in the literature. The hypothesis was that the newly discovered allosteric site would enable them to design better PTP1B allosteric inhibitors. Unfortunately, none of the small molecule fragments bound to the allosteric site exhibited significant inhibitory activity against PTP1B. The authors then attempted tethering compounds to Cys197 and identified compound 2 as an inhibitor using this approach. However, the potency of compound 2 in the tethering assays was still weak, IC50 ca. 50 μm (although it was described as having an IC50 of 7 uM, this is really the concentration of the half-maximal effect of compound 2, not a 50% inhibition of enzyme activity). Overall, while there was enthusiasm for the methodology and scholarship of the study, the lack of a clear pathway to a practical inhibitor for PTP1B reduced support for publication in *eLife*.

*Reviewer #1:*

This work consists of two main results. First, it identified an extensive allosteric network of PTP1B from the concerted conformational changes revealed by room-temperature X-ray crystallography. This represents an important methodology development besides the detailed structural elucidation of PTP1B regulatory mechanism it afforded. Secondly, the work identified several potential sites for allosteric small-molecule inhibition of PTP1B based on a large number of crystal structures from fragment soaking. The results are consistent with known PTP1B allosteric inhibition sites, and revealed a new so-called 197 site, which was further validated by a crystal structure of a small molecule covalently bound there when a Cysteine is introduced (the "tethering" technique) and by the allosteric inhibition of PTP1B confered by the binding. This work fell short of identifying a potent non-covalent small molecule inhibitor for wild-type PTP1B (presumably due to the incapability to synthesis non-off-shelf small molecules), which, of course, is much desired as PTP1B remains an undrugged yet therapeutically highly important target. Notwithstanding, the authors generated a substantial amount of data and new insights into PTP1B concerning its conformational changes and its patterns of interaction with small molecules.

*Reviewer #2:*

This manuscript reports the identification of a putative new allosteric site in PTP1B through a combination of multiple-temperature X-ray crystallography and structure determination of hundreds of individual small-molecule fragments bound to PTP1B. The amount of work described is impressive and the experiments appear to be well executed. The authors noted that there are a number of allosteric inhibitors already described in the literature. The hypothesis was that the newly discovered allosteric site would enable them to design better PTP1B allosteric inhibitors. Unfortunately, none of the small molecule fragments bound to the allosteric site exhibit any inhibitory activity against PTP1B. The authors then attempted tethering which has little impact to the story. The tethering is unlikely to lead to any drug that will have effect on a WT protein. Furthermore, they screened a new library for the tethering screen. Then what was the purpose of the fragment crystal soak screen? Developing a real inhibitor from the hits within the fragment crystal soak screen would have proved their point (PTP1B could be inhibited by binding of a small molecule at this allosteric site) better and been a real step forward in a novel allosteric inhibitor for PTP1B.

*Reviewer #3:*

This work advances our understanding of the subtle collective motions of a prototypical non-receptor protein tyrosine phosphatase, PTP1b. Previous work suggested that the rate of enzymatic catalysis of PTP1b is controlled by the motions of the conserved WPD loop, which has been trapped in different crystals at either open or closed conformations. The authors show the inherent mix of open and closed WPD loop in their reanalysis of the previously deposited PDB structure 1SUG and proceed to demonstrate via multi-temperature crystallography the gradual shift of population towards open WPD as a function of increasing temperature. The collective motion of WPD loop, residues in the alpha3, disorder of alpha7, and residues of L16, match the motions observed upon binding benzbromarone, suggesting the latter stabilizes a pre-existing conformation. The multi-temperature crystallography reveals temperature dependent changes of the 197 site, which might also couple to the WPD motions via Tyr152 and rotation of helix3, thus impacting catalysis. The authors perform a systematic crystallographic fragment screen and identify fragments binding to the benzbromarone site, to L16, and the 197 site. They expand one fragment hit to build a tethered fragment binding to the K197C mutant, for which they show biochemical evidence of inhibition due to tethering and also obtain a high-resolution crystal structure. The crystallography and fragment screening work is solid, and the proposal of a new allosteric site, within reach of a previous site may prove to be important. I'd love to see follow up work with non-tethered inhibitors binding to the same site, but the current work is interesting in its own right.

[Editors’ note: what now follows is the decision letter after the authors submitted for further consideration.]

Thank you for submitting your article "An expanded allosteric network in PTP1B by multitemperature crystallography, fragment screening, and covalent tethering" for consideration by *eLife*. Your article has been reviewed by three peer reviewers, one of whom is a member of our Board of Reviewing Editors and the evaluation has been overseen by Philip Cole as the Senior Editor. The reviewers have opted to remain anonymous.

The reviewers have discussed the reviews with one another and the Reviewing Editor has drafted this decision to help you prepare a revised submission.

Summary:

This manuscript reports on an approach to identify new allosteric networks and allosteric binding pockets in PTP1B phosphatase based on room-temperature X-ray crystallography and crystallographic fragment screening. An extensive allosteric network was identified in agreement with previous reports and, while the work fell short of identifying a potent allosteric PTP1B inhibitor, at least one of the binding pockets is confirmed by covalent "tethering".

Essential revisions:

The reviewers identified a number of concerns in several aspects:

1) Overstatements and over-claims. The manuscript will benefit if the authors focus on the large amount of data (especially the RT crystallography data) they produced and remove and temper their claims.

a) The claim that new pockets are identified by RT crystallography needs qualification. The claim that L16 newly identified is not accurate as it was previously reported by the Loria laboratory. The K197 pocket was also previously reported by Loria lab and Peti lab with a different name. These previous works need to be adequately discussed in this context. The authors need to explain if and how these cryptic binding sites can be revealed from the RT crystallography data without the knowledge a priori.

b) The allosteric network identification is interesting, but a substantial part of the network has been reported in 4 publications (2 by the Loria laboratory and 2 by the Peti laboratory). These results need to be properly acknowledged and discussed. Again, would the network be obvious without the prior knowledge?

c) Some of the added speculative discussion can be misleading and needs to be tempered. For example, the discussion of the BB inhibitors and the corresponding site might be misleading to readers that are not well versed in mechanistic studies or drug discovery. The word "validates" is too strong ("this approach validates many aspects of the recently characterized allosteric network that corresponds to BB inhibitors"), since the referenced paper was largely based on unsubstantiated or weak claims. Even though the present work provides support for some of these speculative ideas it does not validate them.

d) The discussion of flat SAR is misleading: "On the other hand, they may also accommodate different small-molecule variants equally well, such that it is difficult to improve upon inhibition - that is, the structure-activity relationship (SAR) is flat." This argument is questionable as it would be an extreme coincidence to perfectly compensate the binding energetics with corresponding structural rearrangements. The possibility cannot be excluded that the flat SAR is due to an inhibition mechanism that is unrelated to that particular binding site, which is not uncommon occurrence in PTP1b drug discovery efforts.

2) Methodology rigora) How observations are derived from the RT crystallography data in a rigorous way is not well explained throughout the text. Alternative conformations are fit in weak density in published and new crystal structures – however, would it be possible to just place waters in such incomplete density and the statistical analysis of the data improves? What is the cut off for rigor and statistics to identify biologically relevant alternative conformations etc., and, further, that it can be interpreted for the identification of allostery. This issue is crucial to the general applicability of the approach and specifically to the identification of both the allosteric network and the cryptic sites in PTP1B without using prior knowledge.

b) In the fragment screening, many fragments were seen to bind with PTP1B in many locations. How would one know which of these locations are sites for allosteric inhibition if they have not been independently identified previously. It seems in such cases one can only make educated guesses based on druggability estimation of a fragment's binding site and its relation to the structural-functional mechanism of the target's regulation. Or there might be more rigorous ways to approach this issue. This is worth an adequate clarification and discussion.

3) Presentation issuesa) The figures lack continuity and clarity. It seems that in each figure PTP1B is shown from a different orientation, without relaying the orientation to a general overview, which makes it difficult for the reader.

b) The wording in many statements in the manuscript is exaggerated or unnecessarily strong, which needs revision.

c) There are many typos throughout the manuscript.

4) One interesting observation is that the tethering of the compound leads to change in Km? What is the mechanism for this change? A change in the active site? This should be repeated with a set of substrates (e.g. difmup and TK domain substrate peptides with Malachite greenPi capture). Such data may shed new light to the mechanism of allosteric inhibition of PTP1B.

---

## [Author Response]

[Editors’ note: the author responses to the first round of peer review follow.]

All reviewers are complimentary about the technical execution of the work. From that discussion, you identified a need to make the findings about the allosteric network more compelling to appeal to the more critical reviewer. We agree, and the revised manuscript emphasizes the major goal of long-range allosteric control of protein function (by a combination of multitemperature crystallography and chemical biology) and discusses the more limited ramifications for new actionable therapeutic molecules (drug discovery) on this specific protein. Our change to a new title, “An expanded allosteric network in the dynamic enzyme PTP1B revealed by multitemperature crystallography, fragment screening, and covalent tethering”, instead of “New routes for PTP1B allosteric inhibition by multitemperature crystallography, fragment screening and covalent tethering”, summarizes this shift in focus. The revised manuscript better places the allosteric mechanism discovered by multi-temperature crystallography in the context of mutagenesis, recently published NMR work, post-translational modifications, and the partial antagonist effect of the tethered molecule. Moreover, our retrospective evaluation reveals that multi-temperature crystallography plus protein:fragment structures predict the known BB allosteric inhibitor for PTP1B, which demonstrates the potential value of our approach for other targets without any known allosteric inhibitors. By emphasizing the parts of the work that the reviewers were uniformly positive about and placing the parts they were less enthusiastic about in a more self-critical context, we hope that the revised manuscript will be suitable for further consideration at *eLife*.

Reviewer #1:

This work consists of two main results. First, it identified an extensive allosteric network of PTP1B from the concerted conformational changes revealed by room-temperature X-ray crystallography. This represents an important methodology development besides the detailed structural elucidation of PTP1B regulatory mechanism it afforded. Secondly, the work identified several potential sites for allosteric small-molecule inhibition of PTP1B based on a large number of crystal structures from fragment soaking. The results are consistent with known PTP1B allosteric inhibition sites, and revealed a new so-called 197 site, which was further validated by a crystal structure of a small molecule covalently bound there when a Cysteine is introduced (the "tethering" technique) and by the allosteric inhibition of PTP1B confered by the binding.

Thank you for this excellent summary of our work.

This work fell short of identifying a potent non-covalent small molecule inhibitor for wild-type PTP1B (presumably due to the incapability to synthesis non-off-shelf small molecules), which, of course, is much desired as PTP1B remains an undrugged yet therapeutically highly important target. Notwithstanding, the authors generated a substantial amount of data and new insights into PTP1B concerning its conformational changes and its patterns of interaction with small molecules.

As discussed below for reviewer #2, we agree that a potent non-covalent inhibitor for wildtype PTP1B is desirable, but beyond the scope of this manuscript. As reviewer #1 notes, our contribution here is to illuminate the intrinsic allosteric network of the enzyme, map the ligandability of the entire enzyme, and confirm the existence of a functional allosteric linkage from one remote site to the active site. It is unlikely that this site can bind a potent enough molecule without additional interactions, for example with the disordered C-terminal tail, being formed to “cap” the site.

Reviewer #2:

This manuscript reports the identification of a putative new allosteric site in PTP1B through a combination of multiple-temperature X-ray crystallography and structure determination of hundreds of individual small-molecule fragments bound to PTP1B. The amount of work described is impressive and the experiments appear to be well executed.

We thank the reviewer for these kind words about our work.

The authors noted that there are a number of allosteric inhibitors already described in the literature.

There are only two allosteric inhibitors for PTP1B, each with specific limitations. The first type, the BB series, has an IC50 in the μM range and did not reach the clinical stage. The second type, exemplified by MSI-1436, targets the disordered C-terminus – but nonspecifically, at multiple binding sites, and in a pose (or poses) that were not structurally resolved, all of which limit the ability to rationally improve allosteric inhibition. Our work identifies new sites capable of binding diverse small molecules, and we discuss how only some of these sites can likely be made “allosteric”. Our retrospective analysis of the BB binding site and new analysis of the tethering to the 197 site show how linking the knowledge of binding (by fragment screening) and motion (by multitemperature crystallography) could be used to generate candidates for development of new, potent allosteric inhibitors that overcome the limitations of the BB and MSI-1436 series in the future. We have rewritten the corresponding paragraph in the Introduction to better clarify these points, as follows:

“… an allosteric inhibitor that binds to a less-conserved and less-polar surface site could bypass the limitations of active-site inhibitors. Two classes of compounds have been identified that allosterically inhibit PTP1B, although each has limitations. The first class of compounds are based on a benzbromarone (BB) scaffold and inhibit allosterically by binding to the space normally occupied by the regulatory C-terminal α7 helix [10]. Recent work combining mutagenesis, X-ray crystallography, NMR spectroscopy, and molecular dynamics simulations revealed how rotations of the α3 helix and a discrete switch of the catalytic WPD loop are impacted by these BB allosteric inhibitors [11]. Unfortunately, the BB molecules were not successfully translated to the clinic. The second class are natural products, including a molecule called MSI-1436, that bind to multiple sites that primarily involve the disordered C-terminus [4]. However, the binding poses were not structurally resolved, limiting our ability to understand the molecules’ allosteric mechanism and rationally improve their potency. For example, a variant of MSI-1436 had improved inhibition but a distinct response to mutations at the putative binding sites, suggesting an unknown change in mechanism [12]. MSI-1436 passed Phase I clinical trials but was not advanced to Phase II [13]. A new approach to revealing the intrinsic allosteric circuitry of proteins would reveal new opportunities to develop allosteric inhibitors for PTP1B that could overcome the limitations of these existing molecules. Such an approach would additionally set the stage for efforts to dissect allosteric regulatory strategies in other biologically important phospho-signaling proteins.”

The hypothesis was that the newly discovered allosteric site would enable them to design better PTP1B allosteric inhibitors.

Our hypothesis was that the new structural biology methods we apply would reveal sites which can bind small molecules and are functionally coupled to the active site. Indeed, our multitemperature X-ray crystallography to map allosteric connectivity identifies a small subset of binding sites identified by fragment screening, which maps the relative ligandability of the entire protein surface. Remarkably, the methods do converge, in that the sites with the highest number of molecules bound are also the ones identified by multitemperature crystallography. The key result here is that orthogonal methods for mapping how well a site might influence the conformation/activity of the active site, and for mapping the capability of small molecules to bind at that site, converge to the same places. We have added the following text to the Introduction to emphasize this result and its implications:

“To prioritize putative allosteric sites rather than benign binding sites, we focused on the subset of fragment-binding sites that were also conformationally coupled to the active site based on multitemperature crystallography of apo PTP1B. Notably, the sites chosen in this way bound more fragments than did any other sites -- suggesting that conformational heterogeneity may be important for both allostery and ligand binding.”

Importantly, the approach described above is additionally (retrospectively) validated by successfully identifying fragment poses that predict the previously reported BB allosteric inhibitor scaffold (Figure 7A), discovered using independent methods. We have now more strongly emphasized this key point with a new figure, Figure 7—figure supplement 1, which not only details how fragments predict central elements of the BB scaffold, but also provides evidence for potentially fruitful new chemical extensions to that scaffold. We added the following text to the Results section to emphasize this point:

Strikingly, we observed 24 bound fragments in the BB allosteric site (Figure 7A). The poses of many of these fragments overlap portions of the BB scaffold (Figure 7A, Figure 7—figure supplement 1). However, many of them also contain chemical groups that suggestively protrude in new directions from the BB scaffold (Figure 7—figure supplement 1). This retrospective result validates the idea that fragment screening identifies binding sites, and specific fragment poses in those sites, that can be fruitfully exploited for allosteric inhibition.

Unfortunately, none of the small molecule fragments bound to the allosteric site exhibit any inhibitory activity against PTP1B.

The small-molecule fragments have very low affinity due to their small size and lack of any optimization for binding to specific sites in PTP1B, so it was not at all surprising that we did not see any inhibition from them. Indeed, our goal in using fragments was instead to map the bindable surface of the protein. However, the tethering experiments serve a complementary purpose by ultimately confirming a functional link from the allosteric site to the active site. We have clarified these points at several points in the text, including the following in subsection “Validating a functional allosteric linkage with covalent tethering”:

“The small-molecule fragments described above were identified by a naive screen and are not optimized for high-affinity binding to the 197 site or L16 site of PTP1B. Nevertheless, we selected 20 fragments that were deemed to bind in either site during early rounds of iterative PanDDA analysis (see Materials and methods section) (Figure 7—source data 1) and tested whether they have allosteric effects using enzyme activity assays. Unsurprisingly, we did not observe inhibition of enzyme activity by the fragments up to the maximum concentrations we were able to assay due to solubility of the fragments. It is important to note that this is not surprising given the fragments’ relatively simple chemical structures and low affinities (soaking with fragments at 30-150 mM concentrations resulted in observed occupancies of only 10-30% in the crystal structures). However, looking ahead, the dozens of cocrystal structures with small-molecule fragments bound at promising new allosteric sites we have reported here offer a foothold for future medicinal chemistry efforts to design allosteric inhibitors for PTP1B.

Instead, here we focus on an alternative strategy to validate the concept of allosteric inhibition at the 197 site: covalent binding to enhance ligand occupancy. Specifically, we used “Tethering”…”

The authors then attempted tethering which has little impact to the story. The tethering is unlikely to lead to any drug that will have effect on a WT protein.

We agree with the reviewer that converting a covalent tethered compound for a mutant protein to a non-covalent drug for the wildtype protein remains a challenge beyond the scope of this work. However, we disagree that the tethering has little impact: the main goal of the tethering experiment here was to confirm that the allosteric 197 site is functionally coupled to the active site – i.e., to establish the concept of long-range control of protein function from this site. Indeed, our experiments confirm that a local perturbation (the tethered compound) at the 197 site, which is distal from the active site, significantly modulates the activity of the enzyme. This observation is critical for revealing that the 197 site can be thought of as a true allosteric site, rather than a benign, non-functional binding site. Thus, the tethering experiments ultimately confirm the prediction of allosteric coupling between the 197 site and active site that was made based on our multitemperature X-ray analysis of the apo enzyme.

Furthermore, they screened a new library for the tethering screen. Then what was the purpose of the fragment crystal soak screen?

The goal of the fragment screen was to identify places on the protein surface that bind molecules. The tethering screen followed up on the fragment screen on a single site – chosen because it had the highest number of molecules bound (even more than the previously established allosteric inhibitor site). The fragment and tethering libraries are different due to the requirement for a disulfide linkage in the tethering experiment. Thus, the fragments were free to contain essentially any chemical groups, whereas the tethering molecules all were required to feature a disulfide group, and were all screened against a specific mutant (K197C) of the protein. Converting fragment hits from our initial screen to disulfide-containing fragments would have been difficult and slow. It was simpler to rescreen the identified site against the existing disulfide fragment library.

Overall, the fragment screen mapped the ligandability of the entire surface (critically, confirming that the 197 site, BB site, and L16 site were binding hotspots), and the tethering screen identified a compound that tethers well (indicating it binds appreciably) to the 197 site and is functionally linked to activity. Essentially the two approaches (fragments vs. tethering) involve a tradeoff between structural resolution and affinity. By combining both approaches, we have learned that the most bindable site in PTP1B has an inherent allosteric connection to activity.

We have added the following text to the Discussion section to clarify the inherent value of both the fragment and tethering screens:

First, we used a new high-throughput method with small-molecule fragments to map the ligandability of the entire surface of PTP1B with high structural resolution. Although individually these fragments have low affinity, collectively the 110 protein:fragment structures we report reveal overlapping poses with a multitude of precise binding interactions at specific sites in PTP1B (Figure 7A,B,C). Second, we used covalent tethering to probe the functional effects of a high-occupancy ligand at one promising allosteric site. Our work takes advantage of a synergy between fragments and tethering. Fragments bind weakly but allow for the visualization of hundreds of chemical entities bound to distinct sites in a protein. By contrast, tethering does not provide high-throughput structural information, but allows for targeted perturbations to functionally probe a specific site, and additionally can provide lower-throughput structural information about chemical matter that allosterically inhibits a protein variant.

Developing a real inhibitor from the hits within the fragment crystal soak screen would have proved their point (PTP1B could be inhibited by binding of a small molecule at this allosteric site) better and been a real step forward in a novel allosteric inhibitor for PTP1B.

We agree with the reviewer that a “real” non-covalent inhibitor for PTP1B is desirable, as it has been a highly sought-after therapeutic target for many years, but that it is also an extremely high bar for publication.

We have refocused several aspects of the manuscript around what we do think is the major step forward here: revealing an expanded allosteric network in PTP1B via multitemperature crystallography, mapping the ligandability of the entire surface to find potential allosteric sites with high-throughput small-molecule fragment screening and structure determination, and confirming a functional linkage from a new allosteric site to the active site. Although the tethered molecule we report at the allosteric 197 site achieves only modest inhibition, the partial allosteric inhibition we observe is strong evidence that the 197 site represents an important new foothold for allosteric inhibitor development for PTP1B. As described above, we have added text to the Discussion section that comments on the mechanistic basis of this partial antagonism.

Overall, our work introduces a new approach for establishing long-range control of conformation and function. This is a general approach that we have applied to PTP1B. In addition to the novel result of the 197 site, our approach successfully identifies the core scaffold and conformational hypothesis of the previously characterized BB allosteric inhibitors (as noted above, we have added a new figure to the paper to emphasize this key point). Moreover, the unprecedented number of small-molecule:PTP1B cocrystal structures we report offers invaluable opportunities for specific medicinal chemistry follow-up experiments using fragment-based drug design. While we have moved the field toward the goal of developing potent non-covalent inhibitors for PTP1B, our work takes on additional value by being highly generalizable to other systems.

Reviewer #3:

This work advances our understanding of the subtle collective motions of a prototypical non-receptor protein tyrosine phosphatase, PTP1b. Previous work suggested that the rate of enzymatic catalysis of PTP1b is controlled by the motions of the conserved WPD loop, which has been trapped in different crystals at either open or closed conformations. The authors show the inherent mix of open and closed WPD loop in their reanalysis of the previously deposited PDB structure 1SUG and proceed to demonstrate via multi-temperature crystallography the gradual shift of population towards open WPD as a function of increasing temperature. The collective motion of WPD loop, residues in the alpha3, disorder of alpha7, and residues of L16, match the motions observed upon binding benzbromarone, suggesting the latter stabilizes a pre-existing conformation. The multi-temperature crystallography reveals temperature dependent changes of the 197 site, which might also couple to the WPD motions via Tyr152 and rotation of helix3, thus impacting catalysis. The authors perform a systematic crystallographic fragment screen and identify fragments binding to the benzbromarone site, to L16, and the 197 site. They expand one fragment hit to build a tethered fragment binding to the K197C mutant, for which they show biochemical evidence of inhibition due to tethering and also obtain a high-resolution crystal structure.

This is a nice summary of our work from the reviewer.

The crystallography and fragment screening work is solid, and the proposal of a new allosteric site, within reach of a previous site may prove to be important. I'd love to see follow up work with non-tethered inhibitors binding to the same site, but the current work is interesting in its own right.

As mentioned above, we concur that development of a potent non-covalent inhibitor is desirable and is an excellent topic for future work - but we agree with the reviewer here that our current report has quite broad scope and will be of interest to others.

[Editors' note: the author responses to the re-review follow.]

Summary:This manuscript reports on an approach to identify new allosteric networks and allosteric binding pockets in PTP1B phosphatase based on room-temperature X-ray crystallography and crystallographic fragment screening. An extensive allosteric network was identified in agreement with previous reports and, while the work fell short of identifying a potent allosteric PTP1B inhibitor, at least one of the binding pockets is confirmed by covalent "tethering".

We agree with this summary from the reviewers.

Essential revisions:The reviewers identified a number of concerns in several aspects:1) Overstatements and over-claims. The manuscript will benefit if the authors focus on the large amount of data (especially the RT crystallography data) they produced and remove and temper their claims.

We agree with the reviewers that the novel crystallographic data are exciting. To focus on these data and avoid overstating any claims, we have made several changes to the text. For example, we changed this sentence in the Abstract:

“Our results converge on new allosteric sites that are conformationally coupled to the active-site WPD loop, hotspots for fragment binding, and distinct from other recently reported allosteric sites in PTP1B.”

… to the following:

“Our results converge on allosteric sites in PTP1B that are conformationally coupled to the active-site WPD loop and are hotspots for fragment binding.”

We also changed this paragraph in the Introduction:

“Overall, by pinpointing promising functionally linked, ligandable allosteric sites plus structurally resolving dozens of small-molecule binders at those sites, our work opens doors for future development of potent non-covalent small-molecule allosteric inhibitors for PTP1B. More broadly, we illustrate a new approach to characterizing and rationally exploiting coupled conformational heterogeneity to enable long-range control of protein function that is highly generalizable to other systems.”

… to the following:

“Overall, by highlighting promising allosteric sites and ligands that bind to them, our work may aid future development of potent non-covalent small-molecule allosteric inhibitors for PTP1B. More broadly, we illustrate a generalizable approach to characterizing and exploiting coupled conformational heterogeneity to enable long-range control of protein function.”

Other examples of simplifying our claims are scattered throughout the manuscript, some of which are included in responses to other comments below.

a) The claim that new pockets are identified by RT crystallography needs qualification. The claim that L16 newly identified is not accurate as it was previously reported by the Loria laboratory. The K197 pocket was also previously reported by Loria lab and Peti lab with a different name. These previous works need to be adequately discussed in this context. The authors need to explain if and how these cryptic binding sites can be revealed from the RT crystallography data without the knowledge a priori.

We agree with the reviewers that previous work has used NMR, simulation and mutagenesis to identify sites that closely correspond to the 197 site and L16 site that we have identified here. We have therefore removed the word “new” throughout the manuscript when describing these sites. Overall, we feel that our contributions complement these excellent studies by the Peti and Loria groups in a number of ways by revealing atomic details, implicating distinct residues, and identifying small molecules that bind at these sites (as elaborated below).

First, our work using multitemperature crystallography independently identifies similar allosteric sites as did Peti and Loria et al. by other methods including NMR, MD, mutagenesis, and crystallography. We note that, although publication dates may initially suggest otherwise, our work was in fact performed over several years in parallel with (and without knowledge of) those studies; thus, it is highly gratifying to see several studies converge to some similar conclusions. Moreover, our use of multitemperature crystallography affords direct experimental resolution of relevant conformational states of PTP1B throughout the allosteric network in atomic detail. To emphasize this aspect, we have added the following text to the Introduction:

“Our findings provide support for the previously characterized allosteric network in PTP1B that responds to BB inhibitors [11]. Moreover, they reveal extensions of this network, including additional allosteric binding sites that are distinct from the BB site (Figure 1). Similar regions of PTP1B have been implicated as allosteric sites based on mutagenesis coupled with traditional cryogenic X-ray crystallography, molecular dynamics simulations, and NMR spectroscopy [11][15]; here we complement those studies by using multitemperature crystallography to reveal in atomic detail the conformational heterogeneity that allosterically links these sites to the active site.”

We also added this sentence to Results section:

“Our findings thus shed additional light on the mechanism by which loop 11 allosterically communicates with the active site, thus complementing other recent studies using mutagenesis, MD, and NMR to map allostery in PTP1B [11][15].”

Second, there are some minor differences between the residues implicated in allosteric sites between these studies. In the L16 site, we identify residues that the Loria group did not (and the Peti group did not predict the L16 site, as the reviewers mentioned). In the 197 site, we implicate some residues that are not implicated by the Loria group and/or Peti group. These small differences may in part arise from our use of a different technique, multitemperature multiconformer crystallography, which enables us to directly visualize the conformational ensemble for every residue in the protein. To explain this aspect, we have added the following text for the L16 site in the Results section:

“The L16 site was not identified as part of the allosteric network in PTP1B based on a study using mutagenesis, NMR, and MD [11]. However, in a more recent study, several residues lining what we call the L16 site (including Met3, Lys237, and Ser242) were included in a region called “Cluster II”, which was suggested to be a previously unidentified allosteric site based on reciprocal NMR chemical shift perturbations upon mutation of this site or the WPD loop [15]. Our work here using multitemperature crystallography complements these findings by independently identifying this allosteric site using a new methodology, and by revealing in atomic detail how multiple conformational states at the L16 site may aid communication with the active site. Interestingly, a separate approach combining molecular dynamics and machine learning also recently pointed to this area as a potential “cryptic” binding site [31]. Therefore, the L16 site may be not only energetically coupled to the active site, but also capable of forming an under-appreciated small-molecule binding pocket via the conformational heterogeneity we observe.”

We also added the following text for the 197 site in the Results section:

“We next discuss several similarities and a few differences between what we refer to as the 197 site and similar regions implicated by other recent studies of allostery in PTP1B. First, in addition to predicting the L16 site (see above), reciprocal chemical shift changes upon mutation suggested that several residues at both ends of the 197 site (Tyr152, Tyr153, Lys150, Arg105) are part of a region referred to as “Cluster I” that is allosterically linked to the active site [15]. However, that study did not implicate additional key residues on the α3 helix, e.g. Asn193 and Lys197. Second, mutagenesis, NMR, and cryogenic crystallography implicated several elements of our 197 site as being part of the larger allosteric network in PTP1B: loop 11 (including Tyr152 and Tyr153), the α3 helix (especially Asn193), and the α7 helix [11]. Chemical-shift-restrained molecular dynamics simulations further suggested that Tyr152 on loop 11 and Asn193 on the α3 helix have mutually coupled alternative conformations [11]. However, here we highlight additional residues (e.g. Asp148 and Glu157 on the β strands on either end of loop 11) as being conformationally coupled to each other and to the rest of the allosteric network and the active site, and which may collectively form a binding pocket. Therefore, our work accomplishes two things with regards to these existing studies. First, we add support to their findings by reaching similar conclusions using orthogonal methods. Second, we complement the other studies by revealing additional amino acid residues that may play roles in binding and allosteric communication at the 197 site.”

Finally, in contrast to the Peti and Loria work, our study identifies dozens of structures with ligands bound to PTP1B, including dozens in the two key allosteric sites, and one with a molecule that was experimentally demonstrated to inhibit enzyme activity. This is an important, unique contribution relative to previous work. To emphasize this point, we have added text in several places, e.g. the following in the Introduction:

“Our work builds on previous studies of these sites in PTP1B [11][15], which did not report chemical matter that binds to them.”

b) The allosteric network identification is interesting, but a substantial part of the network has been reported in 4 publications (2 by the Loria laboratory and 2 by the Peti laboratory). These results need to be properly acknowledged and discussed. Again, would the network be obvious without the prior knowledge?

Indeed, the allosteric network is obvious from our multitemperature X-ray analysis. We feel it is a credit to all of these research groups that our results agree with each other, despite using very different methods (NMR, MD). See above for further discussion on this point. Moreover, our multitemperature X-ray experiments provide atomistic detail that complements NMR measurements (provided in some cases by simulations, but revealed here directly experimentally). Finally, we add value by providing many structures of PTP1B in complex with small-molecule fragments, which both validate that the featured putative allosteric sites are ligandable and provide starting poses for future drug discovery efforts by others.

c) Some of the added speculative discussion can be misleading and needs to be tempered. For example, the discussion of the BB inhibitors and the corresponding site might be misleading to readers that are not well versed in mechanistic studies or drug discovery. The word "validates" is too strong ("this approach validates many aspects of the recently characterized allosteric network that corresponds to BB inhibitors"), since the referenced paper was largely based on unsubstantiated or weak claims. Even though the present work provides support for some of these speculative ideas it does not validate them.

To avoid misleading the reader by speculating, we have made several changes throughout the manuscript. For example, we have changed this sentence in the Introduction (underscores for emphasis):

“This approach validates many aspects of the recently characterized allosteric network that responds to BB inhibitors…”

… to the following:

“Our findings provide support for the previously hypothesized allosteric network in PTP1B that responds to BB inhibitors…”

d) The discussion of flat SAR is misleading: "On the other hand, they may also accommodate different small-molecule variants equally well, such that it is difficult to improve upon inhibition -- i.e., the structure-activity relationship (SAR) is flat. " This argument is questionable as it would be an extreme coincidence to perfectly compensate the binding energetics with corresponding structural rearrangements. The possibility cannot be excluded that the flat SAR is due to an inhibition mechanism that is unrelated to that particular binding site, which is not uncommon occurrence in PTP1b drug discovery efforts.

We agree with the reviewers that other mechanisms which would also explain the flat SAR should not be excluded. To indicate this, we have added the following to the Discussion section:

“Notably, it is possible that the flat SAR observed at the BB site is due to an inhibition mechanism that is unrelated to the particular binding site, which is not uncommon in PTP1B drug discovery efforts – if this is the case, targeting the 197 site may face similar hurdles.”

2) Methodology rigora) How observations are derived from the RT crystallography data in a rigorous way is not well explained throughout the text. Alternative conformations are fit in weak density in published and new crystal structures – however, would it be possible to just place waters in such incomplete density and the statistical analysis of the data improves? What is the cut off for rigor and statistics to identify biologically relevant alternative conformations etc., and, further, that it can be interpreted for the identification of allostery. This issue is crucial to the general applicability of the approach and specifically to the identification of both the allosteric network and the cryptic sites in PTP1B without using prior knowledge.

We agree that it is important to ensure that our models are not fitting to noise (as described in references Pearce et al., 2017, Lang et al., 2010,van dem Bedem 2009 and Keedy et al., 2014 in particular). Moreover, significant past research has validated the concept of modeling multiple protein states as “alternative conformations”. We have added the following text to the Results section when multiconformer models across temperatures are first introduced to clarify this point:

“Such models are equally good and usually better explanations of the experimental X-ray data [24,25], and have been used to understand many biologically relevant phenomena at protein:water interfaces [26], dynamic enzyme active sites [14,27], and allosteric networks perturbed by mutations [28].”

To further ensure that our multiconformer models for room-temperature and other elevated-temperature datasets in this paper are well-justified, we have now performed an additional analysis in which we removed the alternative conformations, re-refined the new single-conformer models, and compared the resulting R-factors to the multiconformer models’ R-factors. The result is that the multiconformer models’ R-factors are better (lower) in each case. To emphasize this point, we have added this new text to Results section:

“Removing the alternative conformations and re-refining the resulting single-conformer models, either with or without automated solvent placement, yields deteriorated statistics (Table 1—source data 1), which confirms that the multiconformer models are appropriate explanations of the experimental data at each temperature.”

… and a new supplementary table containing these data:

Finally, the shift across temperatures provides an additional validation that protein conformational shifts, rather than water, is responsible for the density. We are working on joint refinement procedures that use these related datasets to increase the data:parameter ratio as a future direction.

b) In the fragment screening, many fragments were seen to bind with PTP1B in many locations. How would one know which of these locations are sites for allosteric inhibition if they have not been independently identified previously. It seems in such cases one can only make educated guesses based on druggability estimation of a fragment's binding site and its relation to the structural-functional mechanism of the target's regulation. Or there might be more rigorous ways to approach this issue. This is worth an adequate clarification and discussion.

Indeed, it is a general challenge in biophysics to predict what sites on a protein surface are good candidates for allosteric inhibition. Even when specific binding sites are known, it is difficult to predict whether small-molecule binding at them will allosterically inhibit protein activity. This is why our multitemperature X-ray analysis coupled to multiconformer modeling for apo PTP1B was so important: it provides a mechanistic way to predict which parts of the protein are capable of transmitting signals to the active site via an allosteric network of coupled conformational motions. Binding sites that abut this allosteric network are more likely to be good sites for allosteric inhibition using small molecules. We have added the following text to the Discussion section to emphasize this key point:

“Many of the 11 binding sites outside the active site are likely to be benign. Importantly, our multitemperature X-ray analysis of the apo protein provides a way to predict which binding sites are instead likely to be allosterically coupled to function. Based on the result that the same small number of sites are both (a) implicated as allosteric by multitemperature X-ray analysis of conformational changes in apo PTP1B and (b) the most ligandable sites from the fragment screen across the entire surface, we conclude that conformational changes may be important for allosteric ligand binding in this protein.”

3) Presentation issuesa) The figures lack continuity and clarity. It seems that in each figure PTP1B is shown from a different orientation, without relaying the orientation to a general overview, which makes it difficult for the reader.

Due to their allosteric nature, the sites we describe are in various areas of PTP1B, which unfortunately does require different viewing orientations in order to see them without obstruction. We apologize if this was confusing. We do note that in the first revision, we included an additional overview figure (now Figure 1) which specifies all the structural sites that are illustrated in subsequent figures, as requested by the original reviewers. We also note that movie versions of several key figures are available, which provide a better 3-dimensional sense. However, to further assist the reader in this regard, we have added text to each structure-based figure that better orients the reader with respect to Figure 1 and/or other relevant overview figures. Because almost all figures are in one of a small number of orientations (“front side” vs. “back side”), but are zoomed in to different regions from these orientations, we feel this is a helpful strategy for guiding the reader. For example, the caption for Figure 4 now includes the following text:

“The viewing orientation in A-D is as in Figure 1B (“back side” of PTP1B), except zoomed in on the loop 16 site (labeled in Figure 1B).”

b) The wording in many statements in the manuscript is exaggerated or unnecessarily strong, which needs revision.

We have adjusted the tone of the manuscript to be more subdued in many instances. For example, we removed the subjective word “beautifully” from this sentence in Results:

“Remarkably, this L16 alternative conformation sampled by apo PTP1B [beautifully] matches the L16 conformation when PTP1B is allosterically inhibited by BB2…”

As another example, we removed the word “exciting” from this sentence in the Discussion:

“Our work reveals [exciting] new opportunities for long-range control of PTP1B function by impinging upon tendrils of this expanded allosteric network with small molecules.”

c) There are many typos throughout the manuscript.

We thank the reviewers for their attention to detail. We have fixed the typos that we observed while carefully revising the manuscript.

4) One interesting observation is that the tethering of the compound leads to change in Km? What is the mechanism for this change? A change in the active site? This should be repeated with a set of substrates (e.g. difmup and TK domain substrate peptides with Malachite greenPi capture). Such data may shed new light to the mechanism of allosteric inhibition of PTP1B.

Although there is a small change in KM, it is not statistically significant and therefore our original manuscript treated it as no change. To clarify this point – and to contrast it with the significant change in Vmax – we have added this Vmax and KM data to the manuscript as panels E and F in Figure 8—figure supplement 2 along with appropriate captions:

We have also added the following text to the body of the manuscript referencing these new panels:

“Michaelis-Menten kinetic analysis of K197C in the presence of **2** (50 µM) showed a statistically significant ~50% reduction in Vmax relative to DMSO treatment, but no significant effect on KM for the pNPP substrate (Figure 8—figure supplement 2C,E,F).”

In addition, we have now performed new experiments with DiFMUP (as suggested by the reviewer) as an alternative substrate. As with pNPP, we observe no statistically significant change in KM (Figure 8—figure supplement 3D), but a reduction in Vmax (Figure 8—figure supplement 3C). The results across both substrates are therefore consistent with a non-competitive allosteric mechanism. We have commented on this result as follows in the “Validating a functional allosteric linkage with covalent tethering” section:

To further profile the inhibitory effect of tethering of 2 on K197C, we assayed the ability of the tethered complex to dephosphorylate the alternative substrate DiFMUP [45]. As with pNPP, the tethered complex was inhibited, with kinetic analysis showing a dramatic reduction in Vmax, but no significant effect on KM (Figure 8—figure supplement 3). These results once again support a partial noncompetitive allosteric mechanism of inhibition.

We have also added a new figure (Figure 8—figure supplement 3) detailing the Vmax and KM data as well as inhibition data with DiFMUP:

In addition to these clarifications and new data, as requested by the reviewer, we have added the following summary of our finding of a non-competitive allosteric mechanism (significant effect on Vmax, and no significant effect on KM):

We note that the non-competitive allosteric mechanism observed suggests that tethering 2 to K197C may shift the protein’s energy landscape in such a way as to alter the kinetics of WPD loop motions. Future work to explore this issue would nicely complement the crystallographic and functional analysis we provide here.